# Parallelization of Non-linear State-Space Models: Scaling Up Liquid-Resistance Liquid-Capacitance Networks for Efficient Sequence Modeling

**Mónika Farsang**
TU Wien
Vienna, Austria
monika.farsang@tuwien.ac.at

**Radu Grosu**
TU Wien
Vienna, Austria
radu.grosu@tuwien.ac.at

## Abstract

We present LrcSSM, a *non-linear* recurrent model that processes long sequences as fast as today's linear state-space layers. By forcing its Jacobian matrix to be diagonal, the full sequence can be solved in parallel, giving $\mathcal{O}(TD)$ computational work and memory and only $\mathcal{O}(\log T)$ sequential depth, for input-sequence length $T$ and a state dimension $D$. Moreover, LrcSSM offers a formal gradient-stability guarantee that other input-varying systems such as Liquid-S4 and Mamba do not provide. Importantly, the diagonal Jacobian structure of our model results in no performance loss compared to the original model with dense Jacobian, and the approach can be generalized to other non-linear recurrent models, demonstrating broader applicability. On a suite of long-range forecasting tasks, we demonstrate that LrcSSM outperforms Transformers, LRU, S5, and Mamba.

## 1 Introduction

With the advent of linear structured state space models (SSMs), more and more architectures have emerged, with increasingly better accuracy and efficiency. While they can be efficiently parallelized, for example with the aid of the parallel scan operator, this is considerably more difficult for traditional, non-linear models. This led to a decreasing interest in non-linear recurrent neural networks (RNNs), although these should arguably capture input correlations in a more refined way through their state.

Fortunately, recent work has shown how to apply the parallel scan operator to non-linear RNNs, by linearizing them in every time step, and by implementing this idea in their DEER framework [27]. Unfortunately, the state-transition matrix (the Jacobian of the model) was not diagonal, which precluded scaling it up to very long sequences. Subsequent work however, succeeded to scale up non-linear RNNs by simply taking the diagonal of the Jacobian matrix, and stabilizing the DEER updates with trust regions. They called this method ELK (evaluating Levenberg-Marquardt via Kalman) [8].

In this paper, we propose an alternative approach to scaling up non-linear RNNs. Instead of disregarding the non-diagonal elements of the model's Jacobian, which might contain important information about the multistep interaction among neurons along feedback loops, we learn a model whose Jacobian matrix is constrained to be diagonal, and whose entries depend on both the current state and current input. As for linear SSMs, our main intuition is that intricate loops of the non-linear SSM state (neural connectivity) matrix can be well summarized by its complex eigenvalues. After all, the synaptic parameters define constant matrices that can be themselves diagonalized.

39th Conference on Neural Information Processing Systems (NeurIPS 2025).

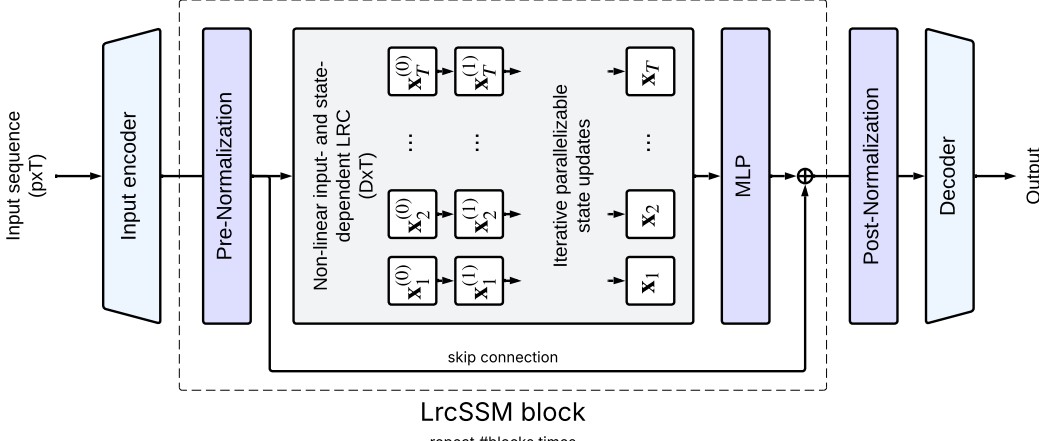

Figure 1: Liquid-Resistance Liquid-Capacitance SSM (LrcSSM) architecture. The input sequence of length $T$ and input dimension $p$ is first passed through an input encoder, followed by a normalization layer. The core component is a non-linear, state-and-input dependent LRC with hidden dimension $D$ and sequence length $T$. This SSM is computed by a parallelizable iterative linearization method. The final state values are then processed by an MLP, with a skip connection added to preserve information flow. The LrcSSM block can be stacked and repeated an arbitrary number of times (we use $2, 4, 6$ layers in our experiments). A post-normalization layer is applied before the output is passed to the decoder, which produces the final output.

To test our idea, we modified and scaled up LRCs (liquid-resistance, liquid-capacitance neural networks), a bio-inspired non-linear RNN [5] that considerably increased LTCs (liquid time-constant networks) accuracy while also decreasing their convergence time [14], by capturing saturation effects in biological neurons, and accounting for the state-and-input dependent nature of their capacitance. Most importantly, we introduce an inherent diagonal form to LRCs, forcing the system's Jacobian to be diagonal. This modification enables exact updates during parallelization - rather than approximations.

Our experimental results on the Heartbeat, SelfRegulationSCP1, SelfRegulationSCP2, EthanolConcentration, MotorImagery, and EigenWorms long-sequence benchmarks show that LrcSSMs are either on par or outperform state-of-the-art SSMs such as NRDE, NCDE, Log-NCDE, LRU, S5, S6, Mamba, LinOSS-IMEX and LinOSS-IM and Transformer variants.

In summary, our main contributions in this paper are the following ones:

- We discuss in detail how to scale LRCs to LrcSSMs having a diagonal non-linear state-and-input dependent state matrix, resulting in an inherently diagonal Jacobian matrix, which allows exact computations via efficient parallelization.

- We demonstrate that LrcSSMs can capture long-horizon tasks in a very competitive fashion on a set of standard benchmarks used to assess accuracy and efficiency of SSMs.

- We show that LrcSSMs consistently outperform many of the state-of-the-art SSMs, including LRU, S5, S6, and Mamba, especially on the EthanolConcentration benchmark.

- We show how our diagonal model approach can be generalized to other non-linear recurrent models, broadening the impact of its design.

## 2   Background

Here we introduce the necessary background for understanding LrcSSMs: Firstly, the bio-inspired non-linear liquid networks - LTCs, STCs, and LRCs - known for their dynamic expressivity. Secondly, the parallelization techniques enabling efficient training of traditionally sequential non-linear models.

---

Our code is available at `https://github.com/MoniFarsang/LrcSSM`

## 2.1 Bio-inspired Liquid Neural Networks

Electrical Equivalent Circuits (EECs) are simplified models defining the dynamic behavior of the membrane potential (MP) of a postsynaptic neuron, as a function of the MP of its presynaptic neurons and external input signals [19, 42]. In ML, EECs with chemical synapses are termed liquid time-constant networks (LTCs) [26, 14]. For a neuron $i$ with $m$ presynaptic neurons of MPs $\mathbf{x}$ and $n$ inputs of value $\mathbf{u}$, the forget conductance $f_i(\mathbf{x}, \mathbf{u})$ and update conductance $z_i(\mathbf{x}, \mathbf{u})$ are defined as:

$$f_i(\mathbf{x}, \mathbf{u}) = \sum_{j=1}^{m+n} g_{ji}^{max} \sigma(a_{ji} y_j + b_{ji}) + g_i^{leak} \tag{1}$$

$$z_i(\mathbf{x}, \mathbf{u}) = \sum_{j=1}^{m+n} k_{ji}^{max} \sigma(a_{ji} y_j + b_{ji}) + g_i^{leak}, \tag{2}$$

where $\mathbf{y} = [\mathbf{x}, \mathbf{u}]$ concatenates the MP (state) of all neurons and the inputs. In Equation (1), $g_{ji}^{max}$ represents the maximum synaptic channel conductance, $a_{ji}$ and $b_{ji}$ parameterize the sigmoidal activation governing channel openness, and $g_i^{leak}$ is the leaking conductance. In Equation (2), $k_{ji}^{max} = g_{ji}^{max} e_{ji}^{rev} / e_i^{leak}$, where $e_{ji}^{rev}$ is the synaptic reversal potential (equilibrium membrane potential) and $e_i^{leak}$ is the leaking potential. Since $g_{ji}^{max} \geq 0$, the sign of $k_{ji}^{max}$ depends on $e_{ji}^{rev} / e_i^{leak}$.

LTC-Equation (4) states that the rate of change of $x_i$ of neuron $i$, is the sum of its forget current $-f_i x_i$ and its update current $z_i e_i^{leak}$. LTCs ignore saturation aspects, which were introduced in saturated LTCs (STCs) in [4]. As $f_i(\mathbf{x}, \mathbf{u})$ is positive, it is saturated with a sigmoid, and as $z_i(\mathbf{x}, \mathbf{u})$ is either positive or negative, it is saturated with a tanh. Saturation is captured in STC-Equation (5).

Finally, both LTCs and STCs assume that the membrane capacitance is constant, and for simplicity equal to 1 in Equations (4)-(5), as the capacitance can be learned jointly with the other parameters. However, this assumption does not hold in biological neurons. In reality, the capacitance has a non-linear dependence on the MP of presynaptic neurons and the external input as they both may cause the neuron to deform [18, 38, 23]. This behavior can be modeled by the following elastance:

$$\sigma(\epsilon_i(\mathbf{x}, \mathbf{u})) = \sigma\left(\sum_{j=1}^{m+n} w_{ji} y_j + v_j\right), \tag{3}$$

where $\mathbf{y} = [\mathbf{x}, \mathbf{u}]$, as before. LRCs incorporate this biological behavior of neurons, by introducing the elastance (which is the reciprocal of the capacitance) as a multiplicative term in LRC-Equation (6).

$$\text{LTC:} \quad \dot{x}_i = \quad -f_i(\mathbf{x}, \mathbf{u}) x_i \quad + \quad z_i(\mathbf{x}, \mathbf{u}) e_i^{leak} \tag{4}$$

$$\text{STC:} \quad \dot{x}_i = -\sigma(f_i(\mathbf{x}, \mathbf{u})) x_i + \tau(z_i(\mathbf{x}, \mathbf{u})) e_i^{leak} \tag{5}$$

$$\text{LRC:} \quad \dot{x}_i = (-\sigma(f_i(\mathbf{x}, \mathbf{u})) x_i + \tau(z_i(\mathbf{x}, \mathbf{u})) e_i^{leak}) \sigma(\epsilon_i(\mathbf{x}, \mathbf{u})) \tag{6}$$

The time constant of LRCs is $\frac{1}{\mathbf{RC}} = \sigma(\mathbf{f}(\mathbf{x}, \mathbf{u})) \sigma(\epsilon(\mathbf{x}, \mathbf{u}))$, which factors into a liquid resistance $\mathbf{R} = 1/\sigma(\mathbf{f}(\mathbf{x}, \mathbf{u}))$ and a liquid capacitance $\mathbf{C} = 1/\sigma(\epsilon(\mathbf{x}, \mathbf{u}))$. While the resistive liquidity is the core of both LTCs and STCs, the capacitive liquidity acts as an additional control in LRCs.

The states $\mathbf{x}$ of LRCs at time $t$ can be computed using the explicit Euler integration scheme as:

$$\text{LRC:} \quad \mathbf{x}_t = \mathbf{x}_{t-1} + \Delta t \dot{\mathbf{x}}_{t-1} \tag{7}$$

## 2.2 Parallelization Techniques

The DEER method [27] formulates next-state computation in non-linear RNNs as a fixed-point problem and solves it using a parallel version of the Newton's method. At each iteration step, DEER linearizes the model. This approximation is widely effective across many domains, and often yields accurate estimates and fast convergence. The main limitation of DEER is the use of a square Jacobian, which does not scale up to long sequences when included in the parallel scan. The second limitation is its numerical instability, which arises from the nature of Newton's method. In particular, the undamped version lacks global convergence guarantees and often diverges in practice [43, 8].

As an improvement, [8] introduces quasi-DEER, which scales DEER by using the diagonal of the Jacobian, only. This is shown to achieve convergence comparable to Newton's method while using less memory and running faster. Nevertheless, quasi-DEER still suffers from limited stability.

To stabilize its convergence, the connection between the Levenberg-Marquardt algorithm and Kalman smoothing is leveraged in the ELK (Evaluating Levenberg-Marquardt with Kalman) algorithm [8]. This stabilisation of the Newton iteration by constraining the step size within a trust region prevents large and numerically unstable updates. As a result, updates are computed using a parallel Kalman smoother, with a running time that is logarithmic in the length of the sequence. Algorithm 1 below, presents these methods [8].

---

**Algorithm 1** DEER/ELK method with optional quasi approximation [8]

---

1: **procedure** PARALLELIZERNN($f, s_0, \text{init\_guess}, \text{tol}, \text{method}, \text{quasi}$)
2:     diff $\leftarrow \infty$
3:     states $\leftarrow$ init\_guess
4:     **while** diff $>$ tol **do**
5:         shifted\_states $\leftarrow [s_0, \text{states}[:-1]]$
6:         $f_s \leftarrow f(\text{shifted\_states})$
7:         $J_s \leftarrow$ GETJACOBIANS($f, \text{shifted\_states}$)
8:         **if** quasi: $J_s \leftarrow$ DIAG($J_s$)
9:         $b_s \leftarrow f_s - J_s \cdot \text{shifted\_states}$
10:       new\_states $\leftarrow$ method($J_s, b_s, \text{states}, s_0$)
11:       diff $\leftarrow \|\text{states} - \text{new\_states}\|_\infty$
12:       states $\leftarrow$ new\_states
13:     **end while**
14:     **return** *states*
15: **end procedure**

---

## 3   Scaling Up Non-linear LRCs

A scalable DEER or ELK approximation, first computes the dense Jacobian of the non-linear RNN, as shown in Line 7 of Algorithm 1, and then extracts its diagonal as shown in Line 8. This results in a quasi approximation of the original DEER/ELK technique, called quasi-DEER and quasi-ELK [8].

**Our Parallelization.** Instead of following this approach, we directly modify the underlying non-linear LRC-Equation (6), such that its Jacobian is diagonal by the model formulation itself. The main idea of this modification is that the state-connectivity submatrices $a^x$, $w^x$ and $g^{max,x}$ are constant parameter matrices that are themselves diagonalizable. Consequently, all cross terms are zeroed out in the LRC through diagonalization. Accordingly, we learn the complex diagonal matrices (vectors) directly, instead.

As a result, our own algorithm is no longer a quasi-approximation, as we do not explicitly remove non-diagonal entries of the Jacobian during parallelization. Instead, we inherently learn their contribution to the dynamics in the model, within the complex eigenvalues of the diagonal. Consequently, Line 8, $J_s \leftarrow \text{Diag}(J_s)$ of Algorithm 1, is not needed anymore, and the update computations become more efficient. In this way, we retain the best of both approaches: A much more precise, more stable, and more scalable, parallelization technique by model design.

### 3.1   Proposed Model

In order to achieve a diagonal Jacobian for the LRCs by model design, we first modify the Equations (1)-(3), by splitting their summation terms into a state-dependent and an input-dependent group, respectively. For the former, we only keep the self-loop synaptic parameters, and zero out all the cross-state synaptic parameters in the associated matrices. For the latter we keep the influence of all external inputs $u$ through their cross-input synaptic parameters, as this part is zeroed out anyway in the Jacobian. To highlight the separation of the terms, we include an extra superscript $x$ for the learnable parameters in the state-dependent part, and the superscript of $u$ for the parameters in the input-dependent part. This separation results in Equations (8)-(10).

As a consequence, instead of keeping cross-synaptic activations, where each individual synapse between neuron $j$ and $i$ has its own $g_{ji}^{max}$, $b_{ji}$ and $k_{ji}^{max}$ as it was in Equation (1) and (2), we now only keep the self-loop neural activations, where the synaptic parameters from the same neuron are equal. Note that instead of the $ij$ indices, we have only the $i$ index for $g^{max}$ and $k^{max}$, and $j$ for $b$ in Equations (8) and (9).

We denote the modified equations of the LRCs with an asterisk. This gives us the following equations for the $f_i^*(x_i, \mathbf{u})$, $z_i^*(x_i, \mathbf{u})$, and $\epsilon_i^*(x_i, \mathbf{u})$ terms:

$$f_i^*(x_i, \mathbf{u}) = \underbrace{g_i^{max,x} \sigma(a_i^x x_i + b_i^x)}_{x_i \text{ state-dependent}} + \underbrace{g_i^{max,u} \sigma(\sum_{j=1}^{n} a_{ji}^u u_j + b_j^u)}_{\mathbf{u} \text{ input-dependent}} + g_i^{leak} \tag{8}$$

$$z_i^*(x_i, \mathbf{u}) = \underbrace{k_i^{max,x} \sigma(a_i^x x_i + b_i^x)}_{x_i \text{ state-dependent}} + \underbrace{k_i^{max,u} \sigma(\sum_{j=1}^{n} a_{ji}^u u_j + b_j^u)}_{\mathbf{u} \text{ input-dependent}} + g_i^{leak} \tag{9}$$

$$\epsilon_i^*(x_i, \mathbf{u}) = \underbrace{w_i^x x_i + v_i^x}_{x_i \text{ state-dependent}} + \underbrace{\sum_{j=1}^{n} w_{ji}^u u_j + v_j^u}_{\mathbf{u} \text{ input-dependent}} \tag{10}$$

LrcSSM: $\quad \dot{x}_i = -\sigma(f_i^*(x_i, \mathbf{u}))\sigma(\epsilon_i^*(x_i, \mathbf{u}))\, x_i + \tau(z_i^*(x_i, \mathbf{u}))\sigma(\epsilon_i^*(x_i, \mathbf{u}))\, e_i^{leak} \tag{11}$

For the final form our proposed LRC model, Equation (11) can be formulated into the form of SSMs, by taking the vectorial form of the states $\mathbf{x}$ of size $m$ and input vector $\mathbf{u}$ of size $n$:

$$\text{LrcSSM:} \quad \dot{\mathbf{x}} = \mathbf{A}(\mathbf{x}, \mathbf{u})\mathbf{x} + \mathbf{b}(\mathbf{x}, \mathbf{u}), \tag{12}$$

where

$$\mathbf{A}(\mathbf{x}, \mathbf{u}) = \text{diag} \begin{bmatrix} -\sigma(f_1^*(x_1, \mathbf{u}))\sigma(\epsilon_1^*(x_1, \mathbf{u})) \\ \dots \\ -\sigma(f_i^*(x_i, \mathbf{u}))\sigma(\epsilon_i^*(x_i, \mathbf{u})) \\ \dots \\ -\sigma(f_m^*(x_m, \mathbf{u}))\sigma(\epsilon_m^*(x_m, \mathbf{u})) \end{bmatrix}, \tag{13}$$

and

$$\mathbf{b}(\mathbf{x}, \mathbf{u}) = \begin{bmatrix} \tau(z_1^*(x_1, \mathbf{u}))\sigma(\epsilon_1^*(x_1, \mathbf{u}))\, e_1^{leak} \\ \dots \\ \tau(z_i^*(x_i, \mathbf{u}))\sigma(\epsilon_i^*(x_i, \mathbf{u}))\, e_i^{leak} \\ \dots \\ \tau(z_m^*(x_m, \mathbf{u}))\sigma(\epsilon_m^*(x_m, \mathbf{u}))\, e_m^{leak} \end{bmatrix}. \tag{14}$$

This reduced diagonal $\mathbf{A}(\mathbf{x}, \mathbf{u})$ form of Equation (13) and the reduced version of $\mathbf{b}(\mathbf{x}, \mathbf{u})$ in Equation (14) results in a diagonal Jacobian matrix which makes the parallelizable iterative state updates exact and efficient, that is, this is not anymore a quasi-approximation of the Jacobian.

## 3.2 Comparison to Linear State Space Models

State-of-the-art time-invariant linear SSMs typically take the following general form:

$$\dot{\mathbf{x}} = \mathbf{A}\mathbf{x} + \mathbf{B}\mathbf{u} \tag{15}$$

$$\mathbf{y} = \mathbf{C}\mathbf{x} + \mathbf{D}\mathbf{u} \tag{16}$$

The main differences between LrcSSM and time-invariant linear SSMs are the following:

- There is no non-linearity in the recurrent state and input update ($\mathbf{A}$ and $\mathbf{B}$, respectively) in time-invariant linear SSMs, which allows them to be parallelized over the time dimension. Here, we investigate non-linear recurrent update and non-linear input update too.

- There are two key aspects of the matrices that state-of-the-art linear SSMs usually follow:

  (1) First, matrix **A** is generally time-invariant (constant), although recent work has introduced an input-dependent variant **A**(**u**) [15, 10]. In our model however, this matrix is both state- and input-dependent, **A**(**x**, **u**). Second, instead of using a traditional **B** matrix that is simply multiplied by the input **u**, we adopt the form **b**(**x**, **u**), allowing the input to have a more embedded influence on the state update.

  (2) Modern linear SSMs typically require a special initialization, such as diagonal plus low rank parameterization of the state matrix of the linear SSMs via higher-order polynomial projection (HiPPO) matrix [11] or only diagonal state matrices with specific parameterization [12, 32]. In our case, we calculate the entries of **A** and **b** from biology-grounded equations of (13) and (14).

### 3.3 Comparison to Liquid-Resistance Liquid-Capacitance Networks (LRCs)

In summary, our approach of LrcSSM differs from LRCs [5] in the following ways:

- Learning in LRCs, like in traditional non-linear RNN models, is inherently sequential. In contrast, we aim for an efficient, parallelizable version in LrcSSMs.

- We modified entries of **A** and **b** to only depend on the self states, rather than on all other states, while still allowing them to depend on the full input. This change yields diagonal Jacobians, exact solutions, and improved efficiency in the update computations.

- While the original LRCs use a single computation layer (a single computation block), we have restructured the LRC architecture into a block-wise design in LrcSSMs, similar to the linear SSM-styled models such as LRU and S5. This design is illustrated in Figure 1.

### 3.4 Theoretical Insights

The LrcSSM architecture enjoys three important theoretical properties. Firstly, by forcing the state-transition matrix of LrcSSMs to be diagonal and learned at every time step, the full sequence can be solved in parallel, giving $\mathscr{O}(TD)$ time and memory and only $\mathscr{O}(\log T)$ sequential depth, where $T$ is the input-sequence length, and $D$ is the state dimension.

Secondly, LrcSSMs offer a formal gradient-stability guarantee that other input-varying systems such as Liquid-S4 and Mamba do not provide. Lastly, because LrcSSM forward and backward passes cost $\Theta(TDL)$ FLOPs, where $L$ is the network depth of the LrcSSM architecture, for its low sequential depth and parameter count $\Theta(DL)$, the model follows the compute-optimal scaling law regime ($\beta \approx 0.42$) recently observed for Mamba, outperforming quadratic-attention Transformers at equal compute while avoiding the memory overhead of FFT-based long convolutions.

The full proof of all these properties is given in Appendix A. In particular we provide details about LrcSSMs stability in A.1 and scalability in A.2.

## 4 Related Work

**Linear Structural State-Space Models.** Since the introduction of S4 [13], linear SSMs rapidly evolved in sequence modeling. S4 used FFT to efficiently solve linear recurrences, and inspired several variants, including S5 [39], which replaced FFT with parallel scans. Liquid-S4 [15] introduced input-dependent state matrices, moving beyond the static structure but relied on FFT. Recent work, such as S6 and Mamba [10], adapted the concept of input-dependency and continued to push linear SSMs to more efficient computation with a hardware-aware parallel algorithm.

**Parallelizing Non-linear Recurrent Neural Networks.** While traditional non-linear RNNs have been favored for their memory efficiency, their major limitation lies in the lack of parallelizability over the sequence length. This has led to the development of parallelizable alternatives, such as [28, 32, 31, 9]. One notable example is the Linear Recurrent Unit (LRU) [32], which uses complex diagonal recurrent state matrices with stable exponential parameterization, achieving comparable performance with SSMs. While LRUs argue that linear recurrence is sufficient, in this work we show that incorporating non-linearity in the transition dynamics can offer significant advantages.

Importantly, these approaches achieve parallelism through entirely new architectures, without addressing how to parallelize existing non-linear RNNs. Techniques like DEER [27] and ELK [8] fill this gap by enabling parallel training and inference for arbitrary non-linear recurrent models.

**Positioning LrcSSM in Recent Advances.** Our LrcSSM aligns with the structured state-space duality (SSD) framework introduced by [3], as its main focus is on designing an RNN that behaves almost like an SSM (diagonalizable and parallelizable). In addition, recent work on parallel state-free inference [33] can be also combined with LrcSSMs to further enhance their efficiency.

## 5    Experiments

We follow the same classification evaluation benchmark proposed in [41] and then used by [37]. These tasks are part of the UEA Multivariate Time Series Classification Archive (UEA-MTSCA). All of these datasets consist of biologically or physiologically grounded time-series data, derived from real-world measurements of dynamic systems, which can be human, animal, or chemical. They capture continuous temporal signals such as neural activity, bodily movements, or spectroscopic readings, making them very well-suited for benchmarking models that need to learn complex temporal dependencies.

### 5.1    Short- and Long-Horizon Sequence Tasks

We compare LrcSSMs against eleven models representing the state of the art for a range of long-sequence tasks. These include the Neural Controlled Differential Equations (NCDE) [22], Neural Rough Differential Equations (NRDE) [30] and Log-NCDE [41], Linear Recurrent Unit (LRU) [32], S5 [39], MAMBA [10], S6 [10], Linear Oscillatory State-Space models with implicit-explicit time integration (LinOSS-IMEX [37]) and with implicit time integration (LINOSS-IM [37]), Transformers [40] and RFormers [29].

We followed the exact same hyperparameter-tuning protocol of [37], using a grid search over the validation accuracy. More details on these experiments are given in Appendix B.2. After fixing the hyperparameters, we compare the average test set accuracy over five different random splits of the dataset. As we are reporting the most of the model results from Rusch et. al [37], we also used the exact same seeds for the dataset splitting as well. When presenting the results, we highlight the top three performing models.

**Short-Horizon Sequence Tasks.** In the first block of Table 1, we report results on datasets with sequence lengths shorter than 1,500 elements. These datasets include the Heartbeat dataset (Heart) [7], which contains heart sound recordings, as well as SelfRegulationSCP1 (SCP1) and SelfRegulationSCP2 (SCP2) [1], which include data on cortical potentials. We report the test accuracy results. We found that our LrcSSM model performed average on these tasks, and we suspect that these datasets lack interesting input correlations.

**Long-Horizon Sequence Tasks.** Next, we focus on the tasks that require learning long-range interactions, especially those with a sequence length above 1,500, up to 18,000. These include the EthanolConcentration dataset (Ethanol) [25], which contains spectroscopic recordings of solutions, the MotorImagery dataset (Motor) [24], which captures data from the motor cortex, and the EigenWorms (Worms) [44] dataset of postural dynamics of the worm *C. elegans*. As shown in the second block of Table 1, our LrcSSM model outperforms all other state-of-the-art SSM methods on the EthanolConcentration task and achieves second-best performance on the MotorImagery and EigenWorms datasets too. We believe that EthanolConcentration contains interesting input correlations, which LrcSSMs can capture through its input- and state-dependence.

**Average Performance Across Datasets.** In the rightmost column of Table 1, we also report the average accuracy across all six datasets considered from the UEA-MTSCA archive. LrcSSM achieved an accuracy of 66.3%, placing it at the forefront alongside the LinOSS-IM model, outperforming all other state-of-the-art models, including LRU, S5, S6, Mamba, and LinOSS-IMEX. The implicit integration scheme of the LinOSS-IM model, seems to have played an important role, and we plan to investigate a similar integration scheme for LrcSSM, too. Our current scheme is just a simple explicit Euler.

Table 1: Test accuracy comparison of different models across relatively *short-horizon* datasets (<1,500) in the left block and *long-horizon* datasets (>1,500) in the middle block. Average test accuracy with standard error (%) across all datasets is in the rightmost block. The performance of the models marked by † is reported from [37], and those with ‡ from [29]. The same hyperparameter tuning protocol and dataset splitting over the same 5 seeds were used.

| | Heart | SCP1 | SCP2 | Ethanol | Motor | Worms | Average |
|---|---|---|---|---|---|---|---|
| Sequence length | 405 | 896 | 1,152 | 1,751 | 3,000 | 17,984 | |
| Input size | 61 | 6 | 7 | 2 | 63 | 6 | |
| #Classes | 2 | 2 | 2 | 4 | 2 | 5 | |
| NRDE[†] | $73.9 \pm 2.6$ | $76.7 \pm 5.6$ | $48.1 \pm 11.4$ | $31.4 \pm 4.5$ | $54.0 \pm 7.8$ | $77.2 \pm 7.1$ | $60.2 \pm 7.7$ |
| NCDE[†] | $68.1 \pm 5.8$ | $80.0 \pm 2.0$ | $49.1 \pm 6.2$ | $22.0 \pm 1.0$ | $51.6 \pm 6.2$ | $62.2 \pm 2.2$ | $55.5 \pm 8.1$ |
| Log-NCDE[†] | $74.2 \pm 2.0$ | $82.1 \pm 1.4$ | $54.0 \pm 2.6$ | $\mathbf{35.9 \pm 6.1}$ | $57.2 \pm 5.6$ | $82.8 \pm 2.7$ | $64.4 \pm 7.6$ |
| LRU[†] | $\mathbf{78.1 \pm 7.6}$ | $84.5 \pm 4.6$ | $47.4 \pm 4.0$ | $23.8 \pm 2.8$ | $51.9 \pm 8.6$ | $85.0 \pm 6.2$ | $61.8 \pm 10.1$ |
| S5[†] | $73.9 \pm 3.1$ | $\mathbf{87.1 \pm 2.1}$ | $\mathbf{55.1 \pm 3.3}$ | $25.6 \pm 3.5$ | $53.0 \pm 3.9$ | $83.9 \pm 4.1$ | $63.1 \pm 9.5$ |
| Mamba[†] | $\mathbf{76.2 \pm 3.8}$ | $80.7 \pm 1.4$ | $48.2 \pm 3.9$ | $27.9 \pm 4.5$ | $47.7 \pm 4.5$ | $70.9 \pm 15.8$ | $58.6 \pm 8.4$ |
| S6[†] | $\mathbf{76.5 \pm 8.3}$ | $82.8 \pm 2.7$ | $49.9 \pm 9.4$ | $26.4 \pm 6.4$ | $51.3 \pm 4.7$ | $85.0 \pm 16.1$ | $62.0 \pm 9.5$ |
| LinOSS-IMEX[†] | $75.5 \pm 4.3$ | $\mathbf{87.5 \pm 4.0}$ | $\mathbf{58.9 \pm 8.1}$ | $29.9 \pm 1.0$ | $\mathbf{57.9 \pm 5.3}$ | $80.0 \pm 2.7$ | $\mathbf{65.0 \pm 8.5}$ |
| LinOSS-IM[†] | $75.8 \pm 3.7$ | $\mathbf{87.8 \pm 2.6}$ | $\mathbf{58.2 \pm 6.9}$ | $29.9 \pm 0.6$ | $\mathbf{60.0 \pm 7.5}$ | $\mathbf{95.0 \pm 4.4}$ | $\mathbf{67.8 \pm 9.7}$ |
| Transformer[‡] | $70.5 \pm 0.1$ | $84.3 \pm 6.3$ | $49.1 \pm 2.5$ | $\mathbf{40.5 \pm 6.3}$ | $50.5 \pm 3.0$ | OOM | $59.0 \pm 8.0$ |
| RFormer[‡] | $72.5 \pm 0.1$ | $81.2 \pm 2.8$ | $52.3 \pm 3.7$ | $34.7 \pm 4.1$ | $55.8 \pm 6.6$ | $\mathbf{90.3 \pm 0.1}$ | $64.5 \pm 8.4$ |
| LrcSSM (Ours) | $72.7 \pm 5.7$ | $85.2 \pm 2.1$ | $53.9 \pm 7.2$ | $\mathbf{36.9 \pm 5.3}$ | $\mathbf{58.6 \pm 3.1}$ | $\mathbf{90.6 \pm 1.4}$ | $\mathbf{66.3 \pm 8.3}$ |

## 5.2 Generalizing the Technique of Diagonal Model Design

Other non-linear RNNs can be transformed into an efficient, parallelizable form by following steps similar to those outlined in Section 3. In Appendix D, we provide a sketch of how this transformation of any non-linear recurrent model can be achieved, and in Table 2, we present results comparing LrcSSM with other non-linear SSM models constructed using the same diagonal model design. In the line of this work, we extend our method to build up other non-linear SSMs from Minimal Gated Units (MGU) [45], Gated Recurrent Units (GRU) [2], and Long Short-Term Memory (LSTM) [16], which we refer to as MguSSM, GruSSM and LstmSSM, respectively. Our experiments show that the LrcSSM model outperforms the other non-linear models across the six classification benchmarks.

Table 2: Experimentation with different non-linear RNN models formulated into efficient SSMs. For these experiments, we use a fixed configuration with input encoding of 64 and 6 blocks of SSMs with 64 units. For each dataset, the average test accuracy and standard deviation (%) are reported, and the last row presents the mean and standard error aggregated across datasets.

| | MguSSM | GruSSM | LstmSSM | LrcSSM (ours) |
|---|---|---|---|---|
| Heart | $74.0 \pm 4.8$ | $\mathbf{75.7 \pm 4.7}$ | $\mathbf{75.0 \pm 3.5}$ | $\mathbf{75.0 \pm 2.6}$ |
| SCP1 | $78.3 \pm 6.6$ | $80.2 \pm 4.2$ | $78.8 \pm 3.1$ | $\mathbf{84.8 \pm 2.8}$ |
| SCP2 | $49.6 \pm 9.9$ | $52.5 \pm 3.3$ | $51.1 \pm 9.5$ | $\mathbf{55.4 \pm 7.7}$ |
| Ethanol | $31.1 \pm 4.2$ | $34.5 \pm 1.9$ | $32.6 \pm 5.6$ | $\mathbf{36.1 \pm 1.1}$ |
| Motor | $\mathbf{56.4 \pm 4.7}$ | $49.6 \pm 7.3$ | $54.3 \pm 3.3$ | $\mathbf{55.7 \pm 4.1}$ |
| Worms | $\mathbf{90.0 \pm 5.2}$ | $86.1 \pm 6.3$ | $82.2 \pm 4.5$ | $85.6 \pm 5.4$ |
| Average | $63.2 \pm 8.8$ | $63.1 \pm 8.4$ | $62.34 \pm 8.0$ | $\mathbf{65.4 \pm 8.0}$ |

## 5.3 Ablation Studies

In Appendix E, we present ablation studies to analyze the contributions of different design choices. First, we compare liquid capacitance (as in LRCs (6)) with constant capacitance (as in STCs (5)) when scaled up to SSMs. Second, we show that simplifying the model to enforce a diagonal Jacobian does not degrade performance, based on experiments against the original full LRC model with dense Jacobians. Finally, we evaluate the impact of input- and state-dependency, as well as the use of real versus complex-valued matrices.

# 6    Discussion

**Competitive Long-Horizon Performance.**    Our experimental evaluations show that the Lrc-SSM model performs moderately well on short-horizon datasets, while demonstrating highly competitive performance on datasets with long input sequences, as shown in Table 1. In those long-sequence tasks, LrcSSMs outperform LRUs, Mamba, and S6, and also achieve better average performance across all datasets.

The only model that LrcSSMs generally does not outperform is the LinOSS-IM model, except on the EthanolConcentration dataset (for both LinOSS-IMEX and LinOSS-IM versions), and on MotorImagery and EigenWorms (in the case of LinOSS-IMEX). This may be attributed to the fact that LinOSS is based on forced linear second-order ODEs, whereas LrcSSMs are built upon LRCs, which are non-linear first-order ODEs. Another possible reason lies in the integration technique: while we were able to outperform the implicit-explicit (IMEX) integration scheme, we did not surpass the fully implicit one (IM) in average test accuracy. This suggests that more sophisticated integration schemes for LrcSSMs (which currently use explicit Euler) may be worth investigating.

**Biological Inspirations in Sequence Modeling.**    We find it particularly interesting that the LinOSS model also exhibits biological relevance, as it models cortical dynamics through harmonic oscillations. In contrast, our approach models information transmission through chemical synapses, which is a different biological phenomenon. The strong performance of both approaches, despite being grounded in different aspects of neuroscience, highlights the significant potential of biologically inspired models as a foundation for future research in sequence modeling.

**Efficient Sequence Modeling with Diagonalized Jacobians.**    In this paper, we focused on the biologically inspired non-linear LRC model, and demonstrated how this model can be made more efficient for long-sequence modeling, by redesigning its underlying state-recurrence matrix **A** and its input-transition vector **b**, such that the resulting Jacobian is a diagonal matrix, for the state-update iterations. This matrix can then be directly used in the parallelizable methods of [8], which gives an exact parallel update, and not an approximation. As we have shown, this approach can also be applied to other non-linear RNNs of interest.

**Limitations.**    As pointed out in Section A.2, this parallelized version holds a good promise towards efficient non-linear RNNs compared to sequential computation costs. Linear SSMs have also the same costs. However, we also have to take into account that LRCs need multiple Newton steps to converge at each iteration, which linear SSMs do not require. The number of iterations depends on the convergence of the state updates, which stops once the difference between the consecutive state updates gets below a defined threshold (see Line 4 of Algorithm 1). The number of iterations per datasets for LrcSSM is shown in Figure 2.

# 7    Conclusion

In this work, we revisited the potential of non-linear RNNs in the era of efficient, scalable linear SSMs. While linear SSMs have seen remarkable success due to their parallelizable structure and computational efficiency, non-linear RNNs have largely been sidelined due to their inherently sequential nature. However, recent advances, particularly the DEER [27] and ELK methods, and their quasi-variants [8] have opened the door to parallelizing non-linear RNNs, thus challenging their long-standing scalability limitation.

Building on these developments, we introduced the liquid-resistance liquid-capacitance non-linear state-space model (LrcSSM), a novel SSM architecture that combines the expressive power of bio-inspired non-linear RNNs with the scalability of modern SSMs. By adapting the parallelization methods and carefully redesigning the internal structure of LRCs, we enable efficient parallel computation by inherently learning diagonal Jacobian matrices, while still preserving the dynamic richness of non-linear state updates in biological neurons. Our design allows for exact parallel updates, rather than relying on quasi-approximations.

Our experiments demonstrate that LrcSSM not only matches but often exceeds the performance of leading linear SSMs such as LRU, S5, S6, and Mamba, particularly in long-horizon sequence

modeling tasks. These results suggest that non-linear-RNN-based SSMs are not only a feasible solution but can also be competitive, offering a promising direction for future research in sequence modeling.

In summary, this work bridges the gap between the expressive flexibility of non-linear dynamics and the computational advantages of parallelism, within the LrcSSMs architecture, opening new pathways for scalable, biologically inspired architectures in modern deep learning.

## Acknowledgements

We thank Ramin Hasani for his feedback and contributions and Daniela Rus for her support in this project. M.F. has received funding from the European Union's Horizon 2020 research and innovation programme under the Marie Skłodowska-Curie grant agreement No 101034277. Experiments were performed on the dataLab cluster at TU Wien, whose support and infrastructure contributed significantly to these research results. We also thank Liquid AI for providing extra computational resources. Finally, we thank the anonymous NeurIPS reviewers for their valuable feedback and comments, which helped to improve our work.

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

# Technical Appendices and Supplementary Material

## A  Theoretical Insights

### A.1  Stability

We analyze a single hidden dimension— because every recurrence is diagonal, all dimensions behave independently and identically. Recall the discrete-time update:

$$x_{t+1} = \lambda_t x_t + b_t, \qquad 0 < \lambda_t \le \rho < 1, \tag{17}$$

where $\rho$ is a user–chosen radius (typically $0.9 - 0.99$) enforced by either the *tanh-clamp* or the *negative–softplus–exponential* parametrisation.

**One step is contractive.**

**Lemma 1** ($\rho$–contraction). *For any $x, y \in \mathbb{R}^D$ we have $\|x_{t+1} - y_{t+1}\|_2 = \|\lambda_t(x - y)\|_2 \le \rho \|x - y\|_2$.*

*Proof.* $\lambda_t$ is diagonal with all entries $\le \rho$, hence its operator (spectral) norm is $\le \rho$; multiplying by it can only shrink Euclidean distances. $\qquad\square$

**Forward states stay bounded.**   Iterating Lemma 1 $t$ times yields

$$\|x_t\|_2 \le \rho^t \|x_0\|_2 + \frac{1 - \rho^t}{1 - \rho} B, \qquad B := \max_{s \le t} \|b_s\|_2. \tag{18}$$

Therefore the hidden state can *never blow up*, irrespective of sequence length.

**Back-propagated gradients never explode.**

**Theorem 1** (Gradient stability). *Let a loss $L$ depend only on the final state $x_T$. Then for any $0 \le \tau < T$*

$$\left\| \nabla_{x_\tau} L \right\|_2 \le \rho^{T-\tau} \left\| \nabla_{x_T} L \right\|_2,$$

*hence the Jacobian product norm is $\le 1$ and cannot explode.*

*Proof.* The Jacobian of one step is $J_t = \lambda_t$, so $\|J_t\|_2 \le \rho$. Back-propagation multiplies $T - \tau$ such Jacobians: $\nabla_{x_\tau} L = J_\tau^\top \cdots J_{T-1}^\top \nabla_{x_T} L$. Sub-multiplicativity of the spectral norm gives the result. $\quad\square$

**Controlled vanishing.**   Because $\rho$ is *tunable*, gradients decay at most geometrically: choosing $\rho \approx 0.99$ keeps long-range signals alive; smaller values add regularisation.

**Deep stacks.**   For $L$ stacked layers with radii $\rho_\ell$ the bound becomes $\left\| \nabla_{x_\tau}^{(\text{layer } L)} L \right\|_2 \le \left( \prod_{\ell=1}^{L} \rho_\ell^{T-\tau} \right) \|\nabla_{x_T} L\|_2$. Keeping every $\rho_\ell$ close to 1 therefore preserves stability in depth.

**How others models handle forward/gradient stability.**   **S4/S6** keep $\mathrm{Re}(A) < 0$ and collapse the recurrence into a single convolution kernel. In this setting, forward activations are bounded and back-propagated Jacobians never appear. **Mamba** re-introduces recurrence via a gate $\sigma(\cdot) \in [0, 1]$; if that gate is clipped the same $\rho$-Lipschitz bound as ours holds, but no proof is given. **LinOSS** discretizes a non-negative diagonal ODE with a symplectic IMEX step, proving both state and gradient norms stay $\le 1$. **Liquid-S4** adds an input term $B u_t$ without clamping the spectrum, so stability relies on empirical eigenvalue clipping. Thus, among truly recurrent models, only LrcSSM (and LinOSS under its specific integrator) enjoy a formal guarantee that *both* forward trajectories and full Jacobian chains remain inside the unit ball.

LrcSSM has a stronger guarantee than Liquid-S4 or Mamba, and—unlike S4-type convolutions, can propagate gradients through actual recurrent steps while remaining provably safe from explosion. This makes training deep, long-sequence stacks straightforward: set $\rho \approx 1$, forget about gradient clipping, and tune $\rho$ itself as a single parameter to trade off memory length versus regularization.

Table 3: Per–layer asymptotic complexity (sequence length $T$, width $D$).

| Architecture | F/B FLOPs | Memory | Parallel depth |
|---|---|---|---|
| Mamba[10] | $\mathcal{O}(TD)$ | $\mathcal{O}(D)$ | $\mathcal{O}(\log T)$ |
| LinOSS[37] | $\mathcal{O}(TD)$ | $\mathcal{O}(D)$ | $\mathcal{O}(\log T)$ |
| Liquid-S4[15] | $\mathcal{O}(TD)$ | $\mathcal{O}(D)$ | $\mathcal{O}(T)$ |
| S4/Hyena[13, 34] | $\mathcal{O}(T\log T D)$ | $\mathcal{O}(T)$ | $\mathcal{O}(\log T)$ |
| Transformer[21] | $\mathcal{O}(T^2 D)$ | $\mathcal{O}(T^2 + TD)$ | $\mathcal{O}(1)$ |
| **LrcSSM** (ours) | $\mathcal{O}(TD)$ | $\mathcal{O}(TD)$ | $\mathcal{O}(\log T)$ |

## A.2 Scalability

Let $T$ denote the input sequence length and $D$ the state dimension. Sequential methods inherently cannot be parallelized, requiring $\mathcal{O}(D)$ memory complexity and $\mathcal{O}(TD^2)$ computational work. Compared to this, the DEER [27] method is parallel but it comes with a major drawback, it requires $\mathcal{O}(TD^2)$ memory complexity and $\mathcal{O}(TD^3)$ computational cost.

The ELK technique introduced in [8] achieves fast and stable parallelization by incorporating diagonal Jacobian computation for scalability. This reduces both memory and computational complexity significantly to $\mathcal{O}(TD)$. Our approach achieves the same complexity — $\mathcal{O}(TD)$ for both memory and computation, thanks to the use of inherently diagonal Jacobians.

Now let's assess formal complexity and compute–optimal scaling laws for LrcSSM:

**Compute, throughput, and memory.** Let FLOPs $\approx c_f B T D L$, be the dominant training cost, where $B$ is the batch size, $T$ the sequence length, $D$ the hidden width, $L$ the network depth, and $c_f$ an architecture–specific constant we define (lower for SSMs and higher for Transformers). The single-GPU throughput (tokens s$^{-1}$ GPU$^{-1}$) is throughput $\approx \frac{TB}{\text{wall-clock time}}$ The *memory footprint* is the sum of peak activations and model parameters.

**Scaling-law** [20, 17]. A *scaling law* is any asymptotic or empirical relation of the form

$$\text{Loss}(C) = A C^{-\beta} + E, \qquad C = \text{compute (FLOPs)}, \quad \beta > 0, \tag{19}$$

or a closed-form complexity identity such as FLOPs $\propto T D$.

Recent large-scale studies like [35] show that $\beta$ depends on the operator's per-token cost: *Dense attention*: $\beta \approx 0.48$–$0.50$ [20]. *Linear-time RNN/SSM (Mamba, Hyena): $\beta \approx 0.42$–$0.45$ in 70 M–7 B runs [10, 34]. *Hybrid (recurrence + sparse attention): $\beta$ can reach 0.41 (MAD pipeline) [35].

Table 3 summarizes the per–layer cost of the main long-sequence architectures in terms of forward/backward FLOPs, peak activation memory, and parallel depth over the sequence length $T$. Because LrcSSM shares the same $\mathcal{O}(TD)$ compute curve as Mamba but with a smaller constant $c_f$ (no low-rank gate, no FFT), we expect it to sit at—or slightly below—the 0.42–0.45 band. The claim is compatible with existing data: Mamba-3B matches a 6-B Transformer at the same FLOPs [10], and LinOSS shows 2× lower NLL than Mamba on 50 k-token sequences at equal compute [37]. Hence, $\beta \approx 0.42$ is a defensible prior for LrcSSM; a hybrid LrcSSM + local-attention block could plausibly move $\beta$ toward 0.41.

**Sequence-length scaling.** For single–GPU throughput $K(T)$, LrcSSM inherits the near-perfect linear behaviour $K(T) \propto T$ of the scan primitive, with practical speed-ups obtainable through width-$w$ windowing and double buffering that saturate L2 cache bandwidth. Liquid-S4 degrades linearly in *latency* because it remains sequential, whereas FFT-based S4/Hyena layers incur $\mathcal{O}(T\log T)$ compute and become memory-bound beyond $T \approx 64$k tokens. Hence, for contexts up to 64k, LrcSSM (and Mamba) are the *compute winners*; at larger $T$ the FFT models may overtake them in raw FLOPs but pay a significant activation cost.

**Sequence-length scaling.** Let $K(T)$ be the wall-clock time for a single forward pass of length $T$ on one GPU. LrcSSM: $K(T) \approx \frac{c}{\text{SMs}} T$ (linear) but can drop to $\approx \frac{c}{\text{SMs}} \frac{T}{w}$ with a width-$w$ scan and double-buffering—near-perfect L2-cache reuse, where SM is the number of CUDA Streaming Multiprocessors on the GPU, and $c$ a hardware-and-kernel–dependent constant (e.g., time per

token per SM). Mamba [10]: same asymptotic, but the fused CUDA kernel shows $\approx 5\times$ higher throughput than a Transformer on 4k tokens; on shorter sequences the constant cost of its scan kernel dominates. S4/Hyena (FFT): $\mathcal{O}(T\log T)$; cross-over with linear methods occurs around $T \approx 8$–16k on A100s—FlashFFTConv reduces the constant $4\times$–$8\times$ [6]. Liquid-S4 [15]: remains sequential; throughput degrades linearly without remedy.

Thus, for $T \leq 64$k, LrcSSM and Mamba are compute winners; beyond 64k, Hyena/S4 win in pure flops but can be memory-bound.

## B   Experimental Details

### B.1   Training Setup

We used A100 GPUs with 80 GB of memory. Training time ranged from less than 1 up to 2-3 hours per data split, depending on the dataset and model. Early stopping was used to prevent overfitting, which varies the training time.

### B.2   Hyperparameters

We performed a grid search over the following set of hyperparameters:

Table 4: Hyperparameter grid. Same values as in [41, 37].

| Parameter name | Value |
|---|---|
| learning rate | $10^{-5}, 10^{-4}, 10^{-3}$ |
| hidden dimension | $16, 64, 128$ |
| state-space dimension | $16, 64, 256$ |
| number of blocks (#blocks) | $2, 4, 6$ |

Using the grid shown in Table 4, we selected the best configuration for each dataset based on the average validation accuracy across five data splits. The splits were generated using the same random seeds as in [37] to ensure full comparability. The final hyperparameters used to report the test accuracies are listed in Table 5.

Table 5: Optimized hyperparameters used for LrcSSM per dataset.

| | lr | hidden dim. | state-space dim. | #blocks | include time |
|---|---|---|---|---|---|
| Heart | $10^{-3}$ | 64 | 64 | 4 | False |
| SCP1 | $10^{-3}$ | 64 | 16 | 2 | False |
| SCP2 | $10^{-3}$ | 128 | 64 | 2 | False |
| Ethanol | $10^{-4}$ | 128 | 16 | 2 | False |
| Motor | $10^{-4}$ | 16 | 16 | 4 | False |
| Worms | $10^{-4}$ | 64 | 16 | 4 | False |

We found that, in general, LrcSSMs benefit from higher learning rates and are not particularly sensitive to the hidden dimension of the encoded input. However, a lower state-space dimension and fewer layers tend to be advantageous.

### B.3   Dataset Sources

The datasets can be downloaded from the following links:

- Short-horizon tasks:
  - Heartbeat (Heart)
  - SelfRegulationSCP1 (SCP1)
  - SelfRegulationSCP2 (SCP2)

- Long-horizon tasks:
  - EthanolConcentration (Ethanol)
  - MotorImagery (Motor)
  - EigenWorms (Worms)

## B.4 Additional Remarks on the Model Design

**Integration Scheme.** As pointed out in Section 6, we used the explicit Euler integration scheme. This is a simple and straightforward solution, but it might be worth investigating more sophisticated and computationally expensive integration methods. In fact, we conducted some preliminary experiments with a hybrid explicit-implicit solver but did not observe any performance improvement, although we did not explore it across the full hyperparameter grid.

**Integration Timestep.** For the integration step, we used a timestep of $\Delta t = 1$ in all our experiments. As [37] investigated different $\Delta t$ values across the datasets and observed no substantial gain in performance, they also continued with $\Delta t = 1$ for all their experiments. However, it might still be worth investigating this in our case as well.

## B.5 Run Times and Number of Iterations

Table 6: Run time in seconds for the considered models for 1000 training steps. Values for the models other than LrcSSM are taken from [37].

|        | NRDE | NCDE  | Log-NCDE | LRU | S5 | Mamba | S6 | LinOSS-IMEX | LinOSS-IM | LrcSSM |
|--------|------|-------|----------|-----|-----|-------|-----|-------------|-----------|--------|
| Heart  | 9539 | 1177  | 826      | 8   | 11  | 34    | 4   | 4           | 7         | 23     |
| SCP1   | 1014 | 973   | 635      | 9   | 17  | 7     | 3   | 42          | 38        | 12     |
| SCP2   | 1404 | 1251  | 583      | 9   | 9   | 32    | 7   | 55          | 22        | 15     |
| Ethanol| 2256 | 2217  | 2056     | 16  | 9   | 255   | 4   | 48          | 8         | 15     |
| Motor  | 7616 | 3778  | 730      | 51  | 16  | 35    | 34  | 128         | 11        | 31     |
| Worms  | 5386 | 24595 | 1956     | 94  | 31  | 122   | 68  | 37          | 90        | 33     |

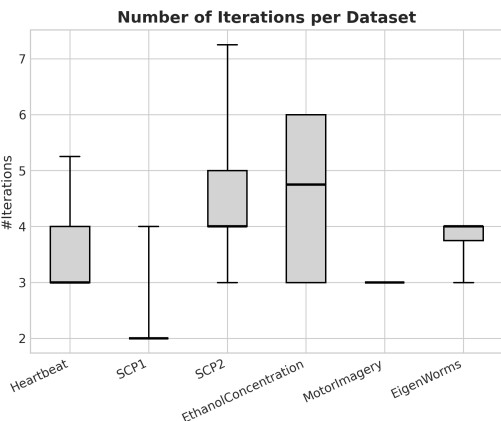

Figure 2: Iterations needed for convergence for LrcSSM on average per dataset.

In Table 6 we present the average run time of the best models for 1000 training steps for each dataset. The values for the models other than LrcSSM are reported from [37]. Figure 2 shows the number of iterations needed for convergence for LrcSSM on average per dataset. As one can see, we do not lose significant runtime due to the extra iterations that are needed for the states to converge through the Newton-steps.

## C  Additional Experiments

We conducted additional experiments on the PPG-DaLiA dataset [36], and report the results (baselines are taken from [37]) in Table 7. We ran the hyperparameter tuning following the same protocol as before. LrcSSM achieves performance comparable to other models, demonstrating its competitiveness on this benchmark.

Table 7: Mean squared error (MSE $\times 10^{-2}$) for different models on PPG-DaLiA. As before, the performance of the models marked by † is reported from [37]. Results are averaged over 5 seeds.

| Model | MSE $\times 10^{-2}$ |
|---|---|
| NRDE[†] | $9.90 \pm 0.97$ |
| NCDE[†] | $13.54 \pm 0.69$ |
| Log-NCDE[†] | $\mathbf{9.56 \pm 0.59}$ |
| LRU[†] | $12.17 \pm 0.49$ |
| S5[†] | $12.63 \pm 1.25$ |
| S6[†] | $12.88 \pm 2.05$ |
| Mamba[†] | $10.65 \pm 2.20$ |
| LinOSS-IMEX[†] | $\mathbf{7.50 \pm 0.46}$ |
| LinOSS-IM[†] | $\mathbf{6.40 \pm 0.23}$ |
| LrcSSM | $10.89 \pm 0.96$ |

## D  Generalized Model Design

Any non-linear RNN can be reformulated by considering its underlying differential equations using an explicit first-order Euler integration scheme with a unit time step. We assume the non-linear RNN has the following general form:

$$\mathbf{x}_t = \mathbf{q}_t(\mathbf{x}_{t-1}, \mathbf{u}_t)\,\mathbf{x}_{t-1} + \mathbf{s}_t(\mathbf{x}_{t-1}, \mathbf{u}_t),$$

where $\mathbf{q}_t = \sigma(\mathbf{f_q}(\mathbf{x}_{t-1}, \mathbf{u}_t))$ and $\mathbf{s}_t = \sigma(\mathbf{f_s}(\mathbf{x}_{t-1}, \mathbf{u}_t))$ are non-linear, potentially state- and input-dependent gated ($\sigma$) functions, which we aim to express it as:

$$\dot{\mathbf{x}} = \mathbf{A}(\mathbf{x}, \mathbf{u})\,\mathbf{x} + \mathbf{b}(\mathbf{x}, \mathbf{u}).$$

To see this, consider:

$$\dot{\mathbf{x}} = (\mathbf{q}(\mathbf{x}, \mathbf{u}) - \mathbf{1})\,\mathbf{x} + \mathbf{s}(\mathbf{x}, \mathbf{u}),$$

which yields (using Euler integration with $\Delta t = 1.0$) the original equation:

$$\mathbf{x}_t = \mathbf{x}_{t-1} + \dot{\mathbf{x}}\Delta t = \mathbf{x}_{t-1} + (\mathbf{q}_t(\mathbf{x}_{t-1}, \mathbf{u}_t) - \mathbf{1})\,\mathbf{x}_{t-1} + \mathbf{s}_t(\mathbf{x}_{t-1}, \mathbf{u}_t) = \mathbf{q}_t(\mathbf{x}_{t-1}, \mathbf{u}_t)\,\mathbf{x}_{t-1} + \mathbf{s}_t(\mathbf{x}_{t-1}, \mathbf{u}_t).$$

Thus, in this general formulation: $\mathbf{A}(\mathbf{x}, \mathbf{u}) = \mathbf{q}(\mathbf{x}, \mathbf{u}) - \mathbf{1}, \quad \mathbf{b}(\mathbf{x}, \mathbf{u}) = \mathbf{s}(\mathbf{x}, \mathbf{u}).$

To obtain the desired efficient model representation, we define:

$$\mathbf{A}(\mathbf{x}, \mathbf{u}) = \mathrm{diag}\left([\sigma(f_{q_1}(x_1, \mathbf{u})) - 1, \ldots, \sigma(f_{q_i}(x_i, \mathbf{u})) - 1, \ldots, \sigma(f_{q_m}(x_m, \mathbf{u})) - 1]\right)$$
$$\mathbf{b}(\mathbf{x}, \mathbf{u}) = [\sigma(f_{s_1}(x_1, \mathbf{u})), \ldots, \sigma(f_{s_i}(x_i, \mathbf{u})), \ldots, \sigma(f_{s_m}(x_m, \mathbf{u}))].$$

### D.1  Example of GRU

The GRU equations are:

$$\mathbf{z}_t = \sigma(\mathbf{W}_z[\mathbf{x}_{t-1}, \mathbf{u}_t] + \mathbf{b}_z) = \sigma(\mathbf{f}_z)$$

$$\mathbf{r}_t = \sigma(\mathbf{W}_r[\mathbf{x}_{t-1}, \mathbf{u}_t] + \mathbf{b}_r) = \sigma(\mathbf{f}_r)$$

$$\mathbf{c}_t = \tanh(\mathbf{W}_h[\mathbf{r}_t\mathbf{x}_{t-1}, \mathbf{u}_t] + \mathbf{b}_h) = \tau(\mathbf{f}_p)$$

$$\mathbf{x}_t = (1 - \mathbf{z}_t)\,\mathbf{x}_{t-1} + \mathbf{z}_t\,\mathbf{c}_t$$

Rewriting the update using the differential form:

$\dot{\mathbf{x}} = \mathbf{z}(-\mathbf{x} + \mathbf{c}) = -\mathbf{z}\mathbf{x} + \mathbf{z}\mathbf{c}$, to check, we get back to the original form (using Euler integration with $\Delta t = 1.0$):

$\mathbf{x}_t = \mathbf{x}_{t-1} + \dot{\mathbf{x}}\Delta t = \mathbf{x}_{t-1} - \mathbf{z}_t\mathbf{x}_{t-1} + \mathbf{z}_t\mathbf{c}_t = (\mathbf{1} - \mathbf{z}_t)\mathbf{x}_{t-1} + \mathbf{z}_t\mathbf{c}_t.$

To express this in the form $\dot{\mathbf{x}} = \mathbf{A}(\mathbf{x}, \mathbf{u})\mathbf{x} + \mathbf{b}(\mathbf{x}, \mathbf{u})$, we set: $\dot{\mathbf{x}} = \mathbf{A}(\mathbf{x}, \mathbf{u})\mathbf{x} + \mathbf{b}(\mathbf{x}, \mathbf{u}) = -\mathbf{z}\mathbf{x} + \mathbf{z}\mathbf{c}$, thus $\mathbf{A}(\mathbf{x}, \mathbf{u}) = -\mathbf{z}$, and $\mathbf{b}(\mathbf{x}, \mathbf{u}) = \mathbf{z}\mathbf{c}$.

Using our proposed formulation for constructing $\mathbf{A}(\mathbf{x}, \mathbf{u})$ and $\mathbf{b}(\mathbf{x}, \mathbf{u})$:

$\mathbf{A}(\mathbf{x}, \mathbf{u}) = \mathrm{diag}\big([-\sigma(f_{z_1}(x_1, \mathbf{u})), \ldots, -\sigma(f_{z_i}(x_i, \mathbf{u})), \ldots, -\sigma(f_{z_m}(x_m, \mathbf{u}))]\big),$

$\mathbf{b}(\mathbf{x}, \mathbf{u}) = [\sigma(f_{z_1}(x_1, \mathbf{u}))\,\tau(f_{p_1}(x_1, \mathbf{u})), \ldots, \sigma(f_{z_i}^*(x_i, \mathbf{u}))\,\tau(f_{p_i}(x_i, \mathbf{u})), \ldots, \sigma(f_{z_m}(x_m, \mathbf{u}))\,\tau(f_{p_m}(x_m, \mathbf{u}))].$

This reformulation results in a diagonal Jacobian matrix, enabling more efficient and exact (rather than quasi) computations for the parallelization method.

## E   Ablation Studies

For the ablation studies below, due to the extensive hyperparameter search required, we fixed the architecture to 6 layers of SSM blocks, each with 64 states, an input encoding dimension of 64, and a learning rate of $10^{-4}$.

**Motivation for LRC and Model with Constant Capacitance.**   We considered models of non-spiking neurons, such as those based on electrical and chemical synapses. Electrical synapses can be modeled using continuous-time recurrent neural networks (CT-RNNs), but they tend to show degraded performance compared to models based on chemical synapses. In such models, part of the transition matrix is fixed, as in many state-space models (SSMs) before, and lacks input or state dependency, which offers nothing novel in this regard.

Instead, we focused on models of chemical synapses, the so-called liquid neural networks, where the term *liquid* refers to the input- and state-dependent dynamics of the transition matrix. These dynamics are non-linear. Within this family, Liquid Time-Constant Networks (LTCs) do not have saturation (or gating) terms in their dynamics of Eq. (4), which is desirable for achieving stable behavior. Fortunately, their saturated (gated) counterpart, Saturated LTCs (STCs), does include such terms as in Eq. (5).

We modified the model of STCs the same way as LRCs, achieving a diagonal Jacobian matrix (without the elastance term indicated in orange in the equations) and ended up with StcSSMs. This model has *constant* capacitance, as assumed in models derived from chemical synapse dynamics.

We compared this model against LrcSSM across the datasets, using a fixed setup. We found that LrcSSM outperforms StcSSM, highlighting the need for the non-linear membrane capacitance as shown in Table 8.

Table 8: Comparison of StcSSM with LrcSSM (proposed model) across datasets.

|         | StcSSM         | LrcSSM         |
|---------|----------------|----------------|
| Heart   | **75.7 ± 5.0** | **75.0 ± 2.6** |
| SCP1    | 78.8 ± 4.3     | **84.8 ± 2.8** |
| SCP2    | 50.4 ± 6.9     | **55.4 ± 7.7** |
| Ethanol | **36.8 ± 4.0** | **36.1 ± 1.1** |
| Motor   | 53.9 ± 4.4     | **55.7 ± 4.1** |
| Worms   | **85.0 ± 4.8** | **85.6 ± 5.4** |
| Average | 63.4 ± 7.8     | **65.4 ± 8.0** |

**Impact of Model Simplification.**   We evaluated the LrcSSM model with a dense Jacobian matrix as well, where $\mathbf{A}$ and $\mathbf{b}$ have all input- and state-dependencies in the model, to compare performance and assess any potential trade-offs. We used the same architectural design as before to ensure a fair comparison.

Table 9: Performance comparison of LrcSSM with the full model resulting in a dense vs. our proposed model with inherently diagonal Jacobian matrix.

|  | LrcSSM-full dense Jacobian | LrcSSM diagonal Jacobian (ours) |
|---|---|---|
| Heart | $72.3 \pm 4.2$ | $\textbf{75.0} \pm \textbf{2.6}$ |
| SCP1 | $83.6 \pm 1.9$ | $\textbf{84.8} \pm \textbf{2.8}$ |
| SCP2 | $49.6 \pm 5.0$ | $\textbf{55.4} \pm \textbf{7.7}$ |
| Ethanol | $33.9 \pm 1.5$ | $\textbf{36.1} \pm \textbf{1.1}$ |
| Motor | $\textbf{55.7} \pm \textbf{5.7}$ | $55.7 \pm 4.1$ |
| Worms | $84.4 \pm 3.8$ | $\textbf{85.6} \pm \textbf{5.4}$ |
| Average | $63.3 \pm 8.3$ | $\textbf{65.4} \pm \textbf{8.0}$ |

These results in Table 9 show that empirically, we do not lose the expressive capabilities of LRCs by constraining the Jacobian matrix to be diagonal. We hypothesize that this is because the strict structure encourages the model to learn more efficient representations, while also enabling exact (rather than approximate) computations due to the design.

**Input- and State-dependency.** We conducted ablation studies to assess the importance of incorporating state-dependency in the state-transition matrix **A** and input-transition vector **b**. As shown in Table 10, the average results indicate that learning both input- and state-dependent transitions yields better performance. We also suggest that future work could treat these dependencies as tunable hyperparameters, as some datasets may benefit from both forms of dependency, while others may perform well with input-dependency alone.

Table 10: Experimentation with different input and state-dependent matrices. Here, we use a fixed configuration with input encoding of 64 and 6 blocks of SSMs with 64 units. For each dataset, the mean and standard deviation are reported, and the last row presents the mean and standard error aggregated across datasets. We found that excluding state dependency from **A** and then from **b** too, downgrades performance on average.

|  | LrcSSM (ours) | LrcSSM | LrcSSM |
|---|---|---|---|
| **A** dependence | $\mathbf{A(x, u)}$ | $\mathbf{A(u)}$ | $\mathbf{A(u)}$ |
| **b** dependence | $\mathbf{b(x, u)}$ | $\mathbf{b(x, u)}$ | $\mathbf{b(u)}$ |
| Heart | $\textbf{75.0} \pm \textbf{2.6}$ | $\textbf{75.0} \pm \textbf{1.8}$ | $73.0 \pm 2.7$ |
| SCP1 | $84.8 \pm 2.8$ | $\textbf{85.0} \pm \textbf{2.9}$ | $83.1 \pm 1.4$ |
| SCP2 | $\textbf{55.4} \pm \textbf{7.7}$ | $49.6 \pm 5.5$ | $51.4 \pm 2.9$ |
| Ethanol | $36.1 \pm 1.1$ | $\textbf{37.6} \pm \textbf{3.9}$ | $34.2 \pm 2.9$ |
| Motor | $55.7 \pm 4.1$ | $\textbf{57.9} \pm \textbf{2.9}$ | $54.3 \pm 6.0$ |
| Worms | $85.6 \pm 5.4$ | $85.0 \pm 5.5$ | $\textbf{86.7} \pm \textbf{5.4}$ |
| Average | $\textbf{65.4} \pm \textbf{8.0}$ | $65.0 \pm 8.0$ | $63.8 \pm 8.4$ |

Please note that results reported here for LrcSSM, do not match the results of the previous tables because we used a fix setup without hyperparameter tuning, to only focus on the importance of state-dependency and changed the underlying matrix **A** and **b** of $\dot{\mathbf{x}} = \mathbf{A(x, u)x} + \mathbf{b(x, u)}$. This results in having even better test accuracies reported here for Heartbeat and SelfRegulationSCP2.

**Complex-valued State-Transition Matrix and Input-Transition Vector.** We also experimented with complex-valued learnable parameters, focusing on those interacting directly with the state **x**. In particular,we experimented with the parameters $g_i^{max,x}$ of $f_i^*(x_i, \mathbf{u})$ and $k_i^{max,x}$ of $z_i^*(x_i, \mathbf{u})$ as defined in Eq.(8) and (9), respectively, as well as their shared sigmoidal channel parameters $a_i^x$ and $b_i^x$. These were gradually converted to complex values, and experiments were conducted using a fixed configuration of 6 SSM blocks, each with 64 state dimensions and 64-dimensional encoded input, and a learning rate of $10^{-4}$. As shown in Table 11, we found no significant performance

gains on average from using complex-valued parameters. As a result, we opted to use real-valued learnable parameters in our main experiments. Nevertheless, we also evaluated the tuned models with their complex-valued counterparts. The only notable improvement occurred on the MotorImagery dataset, where accuracy increased from $54.3 \pm 3.1$ to $58.6 \pm 3.1$. Substituting this result into the average accuracy reported in Table 1 would yield 65.6%, which still ranks our model as the second-best overall.

Table 11: Experimentation with complex valued parameters. Here, we use a fixed configuration with input encoding of 64 and 6 blocks of SSMs with 64 units. For each dataset, the mean and standard deviation are reported, and the last row presents the mean and standard error aggregated across datasets. We found very similar average performance between real-valued and complex-valued parameters.

| | LrcSSM (ours) | LrcSSM with | LrcSSM with | LrcSSM with |
|---|---|---|---|---|
| $g_i^{max,x}$ | $\in \mathbb{R}$ | $\in \mathbb{C}$ | $\in \mathbb{C}$ | $\in \mathbb{C}$ |
| $k_i^{max,x}$ | $\in \mathbb{R}$ | $\in \mathbb{C}$ | $\in \mathbb{C}$ | $\in \mathbb{C}$ |
| $a_i^{x}$ | $\in \mathbb{R}$ | $\in \mathbb{R}$ | $\in \mathbb{C}$ | $\in \mathbb{C}$ |
| $b_i^{x}$ | $\in \mathbb{R}$ | $\in \mathbb{R}$ | $\in \mathbb{R}$ | $\in \mathbb{C}$ |
| Heart | $75.0 \pm 2.6$ | $74.3 \pm 5.2$ | $73.75 \pm 3.2$ | $73.0 \pm 4.0$ |
| SCP1 | $84.8 \pm 2.8$ | $82.9 \pm 2.7$ | $83.1 \pm 4.2$ | $84.8 \pm 2.2$ |
| SCP2 | $55.4 \pm 7.7$ | $50.4 \pm 4.4$ | $53.6 \pm 3.6$ | $58.6 \pm 3.5$ |
| Ethanol | $36.1 \pm 1.1$ | $41.8 \pm 2.1$ | $40.0 \pm 4.5$ | $42.1 \pm 3.6$ |
| Motor | $55.7 \pm 4.1$ | $53.2 \pm 2.6$ | $53.9 \pm 3.5$ | $52.5 \pm 4.3$ |
| Worms | $85.6 \pm 5.4$ | $85.0 \pm 6.5$ | $88.3 \pm 5.7$ | $86.1 \pm 6.3$ |
| Average | $65.4 \pm 8.0$ | $64.6 \pm 7.5$ | $65.4 \pm 7.8$ | $66.2 \pm 7.3$ |

