# OpenReview forum: "Parallelization of Non-linear State-Space Models: Scaling Up Liquid-Resistance Liquid-Capacitance Networks for Efficient Sequence Modeling"
_NeurIPS.cc/2025/Conference — NeurIPS 2025 poster_

### Official Review · Reviewer_hQD2 · 2025-06-30

**Clarity:** 3
**Significance:** 3
**Originality:** 2
**Rating:** 4
**Confidence:** 2

**Summary:**

The authors propose a diagonalisation strategy to speed up the training of nonlinear SSMs. Linearisation is replaced by an adaptive diagonalisation step, which guarantees that the Jacobian of the state-space transition matrices is diagonal without any information loss. The scheme is applied to bio-inspired Liquid-Resistance Liquid Capacitance networks.

**Questions:**

- Is comparing with quadratic-attention Transformers at equal compute the standard practice? Was this principle used to set the sequence lengths in the experiments?
- Why does neglecting the off-diagonal entries of the Jacobian lead to information loss, but enforcing the state-transition matrix to be diagonal does not? What is the difference between the first approach and the inherent diagonalisation proposed on page 2?
- You say that *synaptic parameters define constant matrices*. Is this true for general NSSMs or models with input-dependent transition matrices?
- How strong is the effect of assuming the capacitance to be constant? Is this a standard assumption in the EEC literature? Would the proposed method work if the *capacitance has a nonlinear dependence* (possibly with reduced computational efficiency)?
- Is Euler integration needed at each step? How many iterations were run on average?
- According to the argument that any constant matrix can be diagonalised, it is reasonable to assume that a single state-space matrix is diagonal. When three matrices are involved, it requires the existence of a joint diagonalizer. Is this always the case?

**Ethical Concerns:**

["NO or VERY MINOR ethics concerns only"]

**Final Justification:**

I confirm my positive judgment and will vote to accept the paper.

**Limitations:**

I appreciate that the authors explicitly address the paper's limitations. In my understanding, the work does not have any potential negative societal impact.

**Quality:**

2

**Strengths And Weaknesses:**

**Strengths**
- Bio-inspired models are often praised for their energy efficiency. As traditional deep learning systems become increasingly unsustainable, developing alternative methods is an urgent need.
- The proposed approach demonstrates how enforcing structural task-specific constraints can improve the performance of learning models.
- The idea of an adaptive complex diagonalisation is inspiring.

**Weaknesses**
- The author should spend some more words explaining why they chose Liquid-Resistance Liquid Capacitance networks, why they are relevant, and whether the computational gains are expected to translate straightforwardly to other bio-inspired models.
- The problem with the DEER method is computing the Jacobian of the transition matrix. It is unclear why forcing this by construction is better than approximately updating a full matrix. The authors may explain if this requires a trade-off between speed and accuracy.
- The model requires learning a diagonal matrix at every time step. The authors should comment on its computational costs.
- The models' performance over different datasets has a large variance. Extracting a clear winner from Table 3 is not straightforward. The computational complexity of the methods is not reported.

---

> ### Author Rebuttal · Authors · 2025-07-30
>
> We thank the reviewer for their constructive feedback. Many good points were raised that we address below.
>
> For clarity, we numbered the questions to make them easier to follow. Please note that we moved Questions 2 and 4 to the end, where we also address points raised as weaknesses.
>
> > Q1: Equal compute and sequence lengths in the experiments
>
> Yes, considering equal compute is becoming standard practice for fair comparison. In our experiments, the sequence lengths were determined by the characteristics of the datasets themselves, using their original length.
>
> > Q3: Synaptic parameters are constant matrices, how general...
>
> What we meant is that the synaptic parameters, the learned values $a_{ji}, b_{ji}$, $g_{ji}$ and $k_{ji}$ are not state- or input-dependent, they remain constant after training and define the fixed connectivity between neurons (neuron $j$ and $i$). This is generally true for any NSSM formulation based on this structure.
>
> > Q5: Euler integration and number of iterations
>
> We follow the integration scheme from LRCs, which requires only a single Euler integration step. As a result, the system can be solved immediately without the need for multiple Euler iterations.
>
> > Q6: Matrix diagonalisation, joint diagonalizer
>
> Thank you for bringing this up, we will clarify it in the revision. We explicitly constrain the model to learn diagonal matrices (which can be viewed as vectors). Therefore, the need for a joint diagonalizer does not arise, as the matrices are inherently diagonal by design.
>
> > Weakness 1 and Q4: Motivation for LRC and model with constant capacitance
>
> We considered models of non-spiking neurons, such as those based on electrical and chemical synapses. Electrical synapses can be modeled using continuous-time recurrent neural networks (CT-RNNs), but they tend to show degraded performance compared to models based on chemical synapses. In such models, part of the transition matrix is fixed, as in many state-space models (SSMs) before, and lacks input or state dependency, which offers nothing novel in this regard.
>
> Instead, we focused on models of chemical synapses, the so called liquid neural networks, where the term liquid refers to the input- and state-dependent dynamics of the transition matrix. These dynamics are non-linear. Within this family, Liquid Time-Constant Networks (LTCs) do not have saturation (or gating) terms in their dynamics, which is desirable for achieving stable behavior. Fortunately, their saturated (gated) counterpart, Saturated LTCs (STCs), does include such terms.
>
> We modified the model of STCs the same way as LRCs, achieving a diagonal Jacobian matrix (without the elastance term indicated in orange in the equations) and ended up StcSSMs.
> This model has *constant* capacitance. Models coming from modeling chemical synapses assume this.
>
> We included a comparison of this model against LrcSSM across the datasets, using a fixed setup of 6 layers, 64-dimensional encoded input and 64-dimensional states. We found that LrcSSM outperform StcSSM, highlighting the need for the non-linear membrane capacitance - pointed out by the reviewer. We will include this observation in our ablation section.
>
> | Dataset     | StcSSM       | LrcSSM (proposed model) |
> |-------------|-------------------|--------------------|
> | Heartbeat   | **75.7 ± 5.0**        | **75.0 ± 2.6**         |
> | SCP1        | 78.8 ± 4.3        | **84.8 ± 2.8**         |
> | SCP2        | 50.4 ± 6.9        | **55.4 ± 7.7**         |
> | EthanolConcentration | **36.8 ± 4.0**        | **36.1 ± 1.1**         |
> | MotorImagery | 53.9 ± 4.4        | **55.7 ± 4.1**         |
> | EigenWorms  | **85.0 ± 4.8**        | **85.6 ± 5.4**         |
> | **Average** | 63.4   | **65.4**    |
>
>
> > Weakness 2 and Q2: Comparing diagonal and full matrix results
>
> This is a valuable point that we wanted to address more thoroughly. To this end, we evaluated the LrcSSM model with a dense $A$ matrix as well, to compare performance and assess any potential trade-offs. We used the same architectural design as before to ensure a fair comparison.
>
> | Dataset     | LrcSSM with dense $A$     | LrcSSM (diagonal $A$, proposed) |
> |-------------|------------------|-------------------------|
> | Heartbeat   | 72.3 ± 4.2       | **75.0 ± 2.6**              |
> | SCP1        | 83.6 ± 1.9       | **84.8 ± 2.8**              |
> | SCP2        | 49.6 ± 5.0       | **55.4 ± 7.7**              |
> | EthanolConcentration | 33.9 ± 1.5       | **36.1 ± 1.1**              |
> | MotorImagery | **55.7 ± 5.7**       | **55.7 ± 4.1**              |
> | EigenWorms  | 84.4 ± 3.8       | **85.6 ± 5.4**              |
> | **Average** | 63.3  | **65.4**         |
>
> These results show that empirically, we do not lose the expressive capabilities of LRCs by constraining $A$ to be diagonal. We hypothesize that this is because the strict structure encourages the model to learn more efficient representations, while also enabling exact (rather than approximate) computations due to the design.
>
> This is another important part pointed out by the reviewer, which we will include in the paper.
>
>
>
> > Weakness 3: Computational costs
>
> Let $T$ denote the input sequence length and $D$ the state dimension (i.e. the dimension of the diagonal of the matrix).  Traditional sequential methods are difficult to parallelize and require $O(D)$ memory and $O(T D^2)$ computational work. The ELK technique [1] addresses this by enabling efficient parallelization through diagonal Jacobian computation, reducing both memory and computational complexity to $O(T D)$. Our approach achieves the same complexity: $O(T D)$ for both
> memory and computation. Since our model learns inherently diagonal matrices (i.e., vectors), Jacobians remain diagonal throughout, enabling efficient and scalable computation without additional overhead, making previously quasi-approximations exact.
>
> > Weakness 4: Large average variance in the datasets
>
> We reported the variances in the submission to provide a more complete and transparent view of model performance, but actually, this is something omitted in all prior work using these datasets (only reporting mean values). The high variance is consistent across all models and reflects only the inherent variability in achievable accuracy on these datasets, which shifts the attention away from the more meaningful mean values. We decided now to only keep them for each dataset separately, because it is much more relevant there and shows the reliability of the models across 5 seeds.
>
> References:
>
> [1] Xavier Gonzalez, Andrew Warrington, Jimmy Smith, and Scott Linderman. Towards scalable and stable parallelization of nonlinear rnns. Advances in Neural Information Processing Systems, 37:5817–5849, 2024

---

> > ### Author Response · Authors · 2025-08-05
> > **Follow-up to the reviewer's comments**
> >
> > We hope that our detailed rebuttal has clarified the questions raised by the reviewer.
> >
> > As an update on Weakness 4, and following the discussion with Reviewer dhK1, we will report the standard error as a more appropriate metric for the average results across datasets in our experiments.
> >
> > We look forward to the feedback or questions from the reviewer.

---

> > > ### Comment · Reviewer_hQD2 · 2025-08-05
> > > **Thank you for your answers**
> > >
> > > Many thanks for all your detailed explanations. The additional results on non-diagonal matrices are interesting. Is the better performance of less-flexible models a sign of overfitting? Or does something deeper happen in this specific application?
> > >
> > > I confirm my positive judgment and will vote to accept the paper.

---

> > > > ### Author Response · Authors · 2025-08-05
> > > >
> > > > Thank you for the feedback and for appreciating our work. We also find this observation interesting and plan to explore it further in the future. Our hypothesis is that the effect is not due to overfitting, but rather that the model is being forced to learn better representations, similar to how sparsity can improve efficiency and performance in any kind of ML models.

---

### Official Review · Reviewer_H8n5 · 2025-07-03

**Clarity:** 3
**Significance:** 2
**Originality:** 3
**Rating:** 4
**Confidence:** 4

**Summary:**

The paper introduces LrcSSM, a non-linear state-space model designed for efficient long-sequence modeling. The model is based on the bio-inspired Liquid-Resistance Liquid-Capacitance (LRC) network, a type of non-linear RNN. The core contribution is a modification to the underlying LRC dynamics to enforce that its state-transition Jacobian is inherently diagonal. This "diagonal-by-design" approach allows the model to be parallelized over the sequence length using the iterative ELK (Evaluating Levenberg-Marquardt via Kalman) framework without the need for approximation. The authors demonstrate through experiments on long-sequence classification benchmarks that LrcSSM achieves competitive, and in some cases, state-of-the-art performance compared to linear SSMs like Mamba, S5, and S6.

**Questions:**

**Practical Efficiency:** The central claim of the paper is achieving efficiency comparable to modern SSMs. Could you please provide wall-clock training and inference time comparisons against key baselines like Mamba and S5 on a long-sequence task like EigenWorms? Crucially, how many ELK iterations are typically required for the states to converge during training, and how does this number affect the overall runtime?

**Ethical Concerns:**

["NO or VERY MINOR ethics concerns only"]

**Final Justification:**

Since the reviewers addressed all of my comments, I will update my rating from 3 to 4.

I expect that they will introduce all of this in the new revision of the manuscript, which will strongly improve the paper.

**Limitations:**

Yes, the authors have a discussion section that touches on some limitations, such as the simple integration scheme used. However, the discussion could be significantly strengthened by explicitly addressing the practical computational overhead of the iterative ELK solver compared to the single-pass nature of key SSM baselines. This is a critical limitation of the proposed approach that directly impacts the paper's central claims of efficiency, and it warrants a more prominent and detailed discussion.

**Paper Formatting Concerns:**

None.

**Quality:**

2

**Strengths And Weaknesses:**

**Strengths:**

1.  **Strong Empirical Performance on some Long-Horizon Tasks:** The experimental results on datasets with very long sequences (e.g., EthanolConcentration, EigenWorms) show that the model outperforms several highly-regarded baselines like Mamba and S5 on these specific tasks, suggesting it is effective at capturing certain types of long-range dependencies.
2.  **Formal Stability Guarantees:** The paper provides a theoretical analysis in the appendix showing that the model's gradients are stable and will not explode, a property not formally guaranteed by all input-dependent SSMs. This is a valuable theoretical contribution that adds to the model's robustness.
3.  **Clarity:** The paper is generally well-written and easy to follow. The motivation, background, and proposed method are explained clearly.

**Weaknesses:**

1.  **Potentially Misleading Efficiency Claims:** The primary motivation is to scale non-linear RNNs to be as "efficient" as linear SSMs. However, the model relies on the ELK solver, which is an *iterative* algorithm. Each training step requires multiple forward/backward passes within the solver to converge on the hidden states. In contrast, baselines like Mamba or S5 require only a single, non-iterative parallel scan. The paper presents complexity in terms of a single pass (`O(TD)`) but omits the crucial factor of the number of iterations, `k`. The true cost is closer to `O(k * TD)`. Without wall-clock time comparisons, the claim of comparable efficiency to single-pass SSMs is unsubstantiated and a significant weakness.
2.  **Ambiguous Attribution of Performance Gains:** While the model performs well, it is difficult to disentangle the source of the performance gains. Is it the bio-inspired LRC formulation, the state-and-input dependent non-linearity, or simply that the overall architecture (stacked blocks, normalizations, etc.) is well-tuned for these specific tasks? The paper argues for the superiority of its non-linear approach, but the evidence is not conclusive. For example, the model performs very well on EthanolConcentration but is average on others. A more thorough analysis is needed to isolate the impact of the core non-linear mechanism.
3.  **Limited Significance and Scope:** The proposed method is essentially a clever application of the existing ELK framework to a specific, modified RNN. While successful, it does not represent a new paradigm for sequence modeling. It is an incremental improvement for parallelizing a particular class of non-linear models, and its broader impact on the field may be limited, especially given the practical efficiency concerns.
4.  **Significant Model Simplification:** The "diagonal-by-design" approach requires removing all direct cross-neuron interactions within the state update, a major simplification of the original, more biologically-plausible LRC model. The paper does not adequately discuss the implications of this trade-off in expressivity.

---

> ### Author Rebuttal · Authors · 2025-07-30
>
> We would like to sincerely thank the reviewer for their thoughtful and detailed feedback. Many of the points raised were insightful and important, which we addressed in the following way:
>
> > Weakness 1 and Question on practical efficiency, runtime
>
> Here we report the wall-clock times for the best models for each dataset, in seconds for 1000 training steps:
>
> | Dataset     | NRDE  | NCDE  | Log-NCDE | LRU | S5 | Mamba | S6 | LinOSS-IMEX | LinOSS-IM | LrcSSM (ours) |
> |-------------|-------|-------|----------|-----|----|--------|-----|--------------|------------|--------|
> | Heartbeat   | 9539  | 1177  | 826      | 8   | 11 | 34     | 4   | 4            | 7          | 24     |
> | SCP1        | 1014  | 973   | 635      | 9   | 17 | 7      | 3   | 42           | 38         | 12     |
> | SCP2        | 1404  | 1251  | 583      | 9   | 9  | 32     | 7   | 55           | 22         | 16     |
> | EthanolConcentration | 2256  | 2217  | 2056     | 16  | 9  | 255    | 4   | 48           | 8          | 17     |
> | MotorImagery | 7616  | 3778  | 730      | 51  | 16 | 35     | 34  | 128          | 11         | 29     |
> | EigenWorms  | 5386  | 24595 | 1956     | 94  | 31 | 122    | 68  | 37           | 90         | 33     |
>
> Number of iterations needed for convergence for LrcSSM on average per dataset in the same order: 3.4, 2.1, 4.4, 4.6, 3.0, 3.8.
>
> Results for the other models are from [1]. As one can see, we do not lose significant runtime due to the extra iterations that are needed for the states to converge.
>
> By efficient sequence modeling, we mean that we have taken a step toward scaling nonlinear RNNs to nonlinear state-space models (NSSMs), which can process long sequences in parallel. As the reviewer rightly pointed out, NSSMs require $k$
> iterations to converge to the desired state dynamics, unlike linear SSMs which don't require this. However, compared to traditional nonlinear RNNs, which require sequential computation over the full sequence length $L$, our method overcomes this limitation through parallelization, at the cost of a small number of additional iterations.
>
> In practice, this is highly favorable: the number of iterations $k$ is typically based on experiments very small (between 2 and 4), while sequence lengths $L$ range from 400 up to 18,000. We observed that the number of iterations $k$ is independent of both the sequence length $L$ and the state dimension $D$. Since $k$ is effectively a constant, we believe it is reasonable to omit it from the $O$-notation.
>
> That said, we agree with the reviewer that this should be clearly discussed in the limitations or discussion section of the paper to make our claims more transparent.
>
> > Weakness 2. and 3. Performance gains and broader impact
>
> The reviewer raised a critical point that we aim to address. Although in the submission we only demonstrated how to construct the model from LRC to LrcSSM, this approach can, in theory, be applied to any nonlinear RNNs of interest to obtain its SSM counterpart.
>
> **Sketch of the generalized method:**
>
> Any RNN can be reformulated by considering its underlying differential equations using an explicit first-order Euler integration scheme with a unit time step. We assume the non-linear RNN has the following general form:
>
> $x_t = q_t(x_{t-1},u_t) x_{t-1} + s_t(x_{t-1},u_t)$,
>
> where
> $q_t=\sigma(f_q(x_{t-1},u_t))$ and $s_t=\sigma(f_s(x_{t-1},u_t))$ are non-linear potentially state- and input-dependent gated functions, which we aim to express it as:
>
>  $\dot{x} = A(x,u)x+b(x,u)$.
>
> To see this, consider:
>
> $\dot{x} = (q(x,u)-1)x+s(x,u)$,
>
> which yields (using Euler integration with $\Delta t=1.0$) the original equation:
>
> $x_t = x_{t-1} + \dot{x} \Delta t = x_{t-1} + (q_t (x_{t-1},u_t) -1)  x_{t-1} + s_t(x_{t-1},u_t) = q_t (x_{t-1},u_t)  x_{t-1} + s_t(x_{t-1},u_t) $.
>
> Thus, in this general formulation:
> $A(x,u)=(q(x,u)-1)$,
> $b(x,u) = s(x,u)$.
>
> To obtain the desired efficient representation, we define:
>
> $A(x,u) = \mathrm{diag}([\sigma(f_{q1}(x_1,u)) - 1, \sigma(f_{qi}(x_i,u)) - 1, \ldots, \sigma(f_{qm}(x_m,u)) - 1])$
>
> $b(x,u) = [\sigma(f_{s1}(x_1,u)), \sigma(f_{si}(x_i,u)), ..., \sigma(f_{sm}(x_m,u))]$.
>
>
> **Example: GRU:**
>
> The GRU equations are:
>
> $z_t = \sigma\left(W_z \left[ x_{t-1}, u_t \right] + b_z \right) = \sigma(f_z)$
>
> $r_t = \sigma\left(W_r \left[ x_{t-1}, u_t \right] + b_r \right)= \sigma(f_r) $
>
> $c_t = \tanh\left(W_h \left[ r_t x_{t-1}, u_t \right] + b_h \right) = \tau(f_p) $
>
> $x_t = (1 - z_t)  x_{t-1} + z_t  c_t $
>
> Rewriting the update using the differential form:
>
> $\dot{x} = z(-x + c) = -z x + z c$, to check, we get back to the original form (using Euler integration with $\Delta t=1.0$)
>
> $x_t = x_{t-1} + \dot{x} \Delta t = x_{t-1} + -z_t x_{t-1} +  z_t  c_t   = (1 - z_t) x_{t-1} + z_t c_t $.
>
> To express this in the form $\dot{x} = A(x,u)x+b(x,u)$, we set:
> $\dot{x} = A(x,u)x+b(x,u) =-z x + z c$, thus
> $A(x,u) = -z$, and
> $b(x,u)=z c$.
>
> Using our proposed formulation for constructing $A(x,u)$ and $b(x,u)$:
>
> $A(x,u) = \mathrm{diag}([-\sigma(f_{z1}(x_1,u)), -\sigma(f_{zi}(x_i,u)), ..., -\sigma(f_{zm}(x_m,u))]$,
>
> $b(x,u) = [\sigma(f_{z1}(x_1,u))\tau(f_{p1}(x_1,u)), \sigma(f^*_{zi}(x_i,u))\tau(f_{pi}(x_i,u)), ..., \sigma(f_{zm}(x_m,u))\tau(f_{pm}(x_m,u))]$.
>
> This reformulation results in a diagonal Jacobian matrix, enabling more efficient and exact (rather than quasi) computations for ELK.
>
> In the line of this work, we extend our method to build up other nonlinear SSMs from Minimal Gated Units (MGU) [2], Gated Recurrent Units (GRU) [3], and LSTM [4], which we refer to as MguSSM, GruSSM and LstmSSM, respectively. We report their performance in the following table:
>
> Test accuracy across these *non-linear* models:
>
> | Dataset     | MguSSM      | GruSSM       | LstmSSM      | LrcSSM (ours)      |
> |-------------|---------------------|----------------------|----------------------|----------------------|
> | Heartbeat   | 74.00 ± 4.78        | **75.67 ± 4.67**         | **75.00 ± 3.50**        | **75.0 ± 2.6**           |
> | SCP1        | 78.33 ± 6.62        | 80.24 ± 4.16         | 78.81 ± 3.05         | **84.8 ± 2.8**           |
> | SCP2        | 49.64 ± 9.87        | 52.50 ± 3.31         | 51.07 ± 9.49         | **55.4 ± 7.7**          |
> | EthanolConcentration | 31.05 ± 4.21        | 34.47 ± 1.93         | 32.63 ± 5.61         | **36.1 ± 1.1**           |
> | MotorImagery | **56.43 ± 4.74**        | 49.64 ± 7.27         | 54.29 ± 3.31         | **55.7 ± 4.1**           |
> | EigenWorms  | **90.00 ± 5.15**        | 86.11 ± 6.33         | 82.22 ± 4.51         | 85.6 ± 5.4           |
> | **Average** | 63.2        | 63.1         | 62.34        | **65.4**         |
>
> Here, similar to our ablation studies, we used 6 layers with a 64-dimensional input encoding and 64-dimensional state. The results show that LrcSSM outperforms the other NSSMs built from MGU, GRU and LSTM using the same architecture.
>
> Note: The stacked blocks (2,4 or 6 are part of the hyperparameter search grid) and the normalization blocks are also present in other models reported too.
>
> Overall, with this, we show the performance gains in the same architecture across different non-linear models and we also show the broader impact of our paper, as this diagonalization technique can be applied to many RNNs.
>
>
> > Weakness 4: Model simplification
>
> Again, this is a valuable point that we wanted to address more thoroughly. To this end, we evaluated the LrcSSM model with a dense $A$ matrix as well, to compare performance and assess any potential trade-offs. We used the same architectural design as before to ensure a fair comparison.
>
> | Dataset     | LrcSSM with dense $A$     | LrcSSM (diagonal $A$, proposed) |
> |-------------|------------------|-------------------------|
> | Heartbeat   | 72.3 ± 4.2       | **75.0 ± 2.6**              |
> | SCP1        | 83.6 ± 1.9       | **84.8 ± 2.8**              |
> | SCP2        | 49.6 ± 5.0       | **55.4 ± 7.7**              |
> | EthanolConcentration | 33.9 ± 1.5       | **36.1 ± 1.1**              |
> | MotorImagery | **55.7 ± 5.7**       | **55.7 ± 4.1**              |
> | EigenWorms  | 84.4 ± 3.8       | **85.6 ± 5.4**              |
> | **Average** | 63.3  | **65.4**         |
>
> These results show that empirically, we do not lose the expressive capabilities of LRCs by constraining $A$ to be diagonal. We hypothesize that this is because the strict structure encourages the model to learn more efficient representations, while also enabling exact (rather than approximate) computations due to the design.
>
> References:
>
> [1] T. Konstantin Rusch and Daniela Rus. "Oscillatory State-Space Models". Thirteenth International Conference on Learning Representations (2025).
>
> [2] Zhou, Guoxiang, Jianxin Wu, Chen-Lin Zhang and Zhi-Hua Zhou. “Minimal gated unit for recurrent neural networks.” International Journal of Automation and Computing 13 (2016): 226 - 234.
>
> [3] Cho, Kyunghyun, Bart van Merrienboer, Çaglar Gülçehre, Dzmitry Bahdanau, Fethi Bougares, Holger Schwenk and Yoshua Bengio. “Learning Phrase Representations using RNN Encoder–Decoder for Statistical Machine Translation.” Conference on Empirical Methods in Natural Language Processing (2014).
>
> [4] S. Hochreiter and J. Schmidhuber, "Long Short-Term Memory," in Neural Computation, vol. 9, no. 8, pp. 1735-1780, 15 Nov. 1997, doi: 10.1162/neco.1997.9.8.1735.

---

> > ### Comment · Reviewer_H8n5 · 2025-08-02
> > **[Rating update] All comments addressed**
> >
> > Since the reviewers addressed all of my comments, I will update my rating from 3 to 4.
> >
> > I expect that they will introduce all of this in the new revision of the manuscript, which will strongly improve the paper.

---

> > > ### Author Response · Authors · 2025-08-03
> > >
> > > Thank you for the updated rating and for recognizing our efforts to address all your comments. We appreciate your constructive feedback, especially regarding model simplification and the significance of our work. We are glad that we could address these points and agree that incorporating them will significantly strengthen the paper.

---

### Official Review · Reviewer_3zsp · 2025-07-03

**Clarity:** 3
**Significance:** 2
**Originality:** 3
**Rating:** 4
**Confidence:** 3

**Summary:**

The paper presents a variation of the liquid-resistance, liquid capacitance recurrent neural network. A closely related method, which is ELK (and their algorithm 1) involves the computation of a Jacobian (on line 7) followed by the extraction of the diagonal of that Jacobian, which is done to facilitate a computationally tractable approximation. The contribution here involves modifying the underlying recurrent neural network architecture in such a way as to obtain a Jacobian that is diagonal by construction. This makes things tractable while avoiding the approximation. The resulting algorithm is demonstrated to be highly competitive in terms of accuracy, while being argued (if not quantitatively) to be efficient. There is also an interesting theoretical stability result that is claimed not to exist for the related methods.

**Questions:**

appendix. Please move the definition forward or otherwise drop it.
Q2) Is it possible to compute the diagonal of the jacobian efficiently without simply computing the full dense jacobian first? I am referring here to the ELK algorithm, not the proposed method which obviates the question by having a diagonal JAcobian by construction.
Q3) Can you share some timings? There are theoretical big O complexities in the appendix, but as these are not actually better than the related methods, and since efficiency is a big point in the paper, it would be helpful to get an idea of run times.
Q4) Unless I missed it, there are no discrete sequences here. Is this possible? Why isn’t it included?
Q5) (related to Q4) Is it possible to compare with a transformer?
Q6) Why not include the analogous discretizations for STC and LRC in equation (7)? Apologies if I am missing something here.
Q7) Why did you not use the PPG-Dalia dataset from [32]? The omission is odd given you use everything else.
Q8) In figure 1, wouldn’t it make sense to move the pre-normalization block outside the dotted box?
Q9) Why only 2, 4 and 6 layers? This seems super shallow compared to transformers.
Q10) What was the actual objective / loss function for training?

Minor points:
M1 Bad comma on line 97
M2 Save space by transposing table 3?
M3 Use \mathrm for the max in (1) and (2) and similarly throughout the paper.
M4 Line 284 - quasi approximation is unclear.

**Ethical Concerns:**

["NO or VERY MINOR ethics concerns only"]

**Final Justification:**

I have considered everything already, thanks.

**Limitations:**

yes

**Quality:**

3

**Strengths And Weaknesses:**

Strengths:

The paper attacks an important problem, which is long range dependencies and efficient sequence modelling. The contribution is rather elegant - avoiding an approximation by cleverly designing the transition model to not need that approximation, namely the diagonal approximation of the aforementioned Jacobian. It is also cool to see a theoretical stability result, although I have not verified this result in detail (which is in the appendix).

The empirical results are pretty strong. Given that table 1 was created in [32] it is impressive to match it, since presumably that work chose a setup that is good for their method.

Weakness:

The actual timing gains are not demonstrated in detail. I understand that this is difficult when a lot depends on the details of the implementation, and it is not reasonable to reimplement all related methods.

It would be good to at least include the theoretical statement in the main paper, if not the proof.

There are a number of potential minor weaknesses in the questions section, though the authors may please resolve my misunderstanding for a number of them if possible.

---

> ### Author Rebuttal · Authors · 2025-07-30
>
> We thank the reviewer for their thoughtful feedback, which has helped us improve our work.
>
> See our responses below:
>
> > appendix. Please move the definition forward or otherwise drop it.
>
> Noted. Thank you.
>
> > Q2) Efficient computation of the diagonal of the jacobian
>
> No, unfortunately, it is not straightforward.
>
> > Q3) Run times
>
> Here are our wall-clock times for the best models for each dataset, in seconds for 1000 training steps:
>
> | Dataset     | NRDE  | NCDE  | Log-NCDE | LRU | S5 | Mamba | S6 | LinOSS-IMEX | LinOSS-IM | LrcSSM (ours) |
> |-------------|-------|-------|----------|-----|----|--------|-----|--------------|------------|--------|
> | Heartbeat   | 9539  | 1177  | 826      | 8   | 11 | 34     | 4   | 4            | 7          | 24     |
> | SCP1        | 1014  | 973   | 635      | 9   | 17 | 7      | 3   | 42           | 38         | 12     |
> | SCP2        | 1404  | 1251  | 583      | 9   | 9  | 32     | 7   | 55           | 22         | 16     |
> | EthanolConcentration | 2256  | 2217  | 2056     | 16  | 9  | 255    | 4   | 48           | 8          | 17     |
> | MotorImagery | 7616  | 3778  | 730      | 51  | 16 | 35     | 34  | 128          | 11         | 29     |
> | EigenWorms  | 5386  | 24595 | 1956     | 94  | 31 | 122    | 68  | 37           | 90         | 33     |
>
> Results for the other models are from [1]. We would like to point out that our efficiency claims are *not against* other linear models, because they are really efficient. We aimed to bring up *non-linear* models to the same/comparable efficiency.
> Iterations needed for convergence for LrcSSM on average per dataset in the same order: 3.4, 2.1, 4.4, 4.6, 3.0, 3.8.
> To make it explicit, this number is 1.0 for all other models across all datasets.
> As one can see, we do not gain significant runtime due to the extra iterations that are needed for the states to converge.
>
> > Q4) Discrete sequences
>
> We are not sure if we get the question. The datasets we are using are discrete sequences.
>
> > Q5) Transformers
>
> Good point. Before, we missed a recent publication [2] from last NeurIPS, where Transformers and Rough Transformers (RFormer) were evaluated on the same datasets. We report their results [2] here, which show that LrcSSM achieves better performance than these Transformer-based models. We will incorporate these new model comparisons into our paper.
>
> | Dataset    | Transformer       | RFormer          | LrcSSM (ours)     |
> |------------|-------------------|------------------|-------------------|
> | Heartbeat  | 70.5 ± 0.1        | **72.5 ± 0.1**       | **72.7 ± 5.7**        |
> | SCP1       | 84.3 ± 6.3        | 81.2 ± 2.8       | **85.2 ± 2.1**        |
> | SCP2       | 49.1 ± 2.5        | 52.3 ± 3.7       | **53.9 ± 7.2**       |
> | EthanolConcentration | **40.5 ± 6.3**        | 34.7 ± 4.1       | 36.9 ± 5.3        |
> | MotorImagery | 50.5 ± 3.0        | 55.8 ± 6.6       | **58.6 ± 3.1**        |
> | EigenWorms | OOM               | **90.3 ± 0.1**       | **90.6 ± 1.4**        |
> | **Average**| 59.0      | 64.5      | **66.3**       |
>
> > Q6) Analogous discretizations for STC
>
> Indeed, the same discretization approach can be applied to STCs. One might naturally ask why a model similar to LRC was not constructed for STCs. To address this, we present an alternative formulation of StcSSM, following the same steps used to achieve a diagonal Jacobian matrix (without the elastance term indicated in orange), in order to compare the performance of these two bio-inspired models.
>
> Test accuracy across the same model architecture of 6 layers, 64-dimensional input encoding and 64-dimensional states:
>
> | Dataset     | StcSSM (also ours, but not the main proposed model)        | LrcSSM (ours)  |
> |-------------|--------------------|----------------------|
> | Heartbeat   | **75.7 ± 5.0**         | **75.0 ± 2.6**           |
> | SCP1        | 78.8 ± 4.3         | **84.8 ± 2.8**           |
> | SCP2        | 50.4 ± 6.9         | **55.4 ± 7.7**           |
> | EthanolConcentration | **36.8 ± 4.0**         | **36.1 ± 1.1**           |
> | MotorImagery | 53.9 ± 4.4         |**55.7 ± 4.1**           |
> | EigenWorms  | **85.0 ± 4.8**         | **85.6 ± 5.4**           |
> | **Average** | 63.4        | **65.4**          |
>
> These results show that our originally proposed LrcSSM model performs better than StcSSM, and this performance gain comes from the extra expressiveness of the underlying LRC model, which has an input- and state-dependent membrane capacitance in its bio-inspired model, which is a constant term in the case of the STC model.
>
> > Q7) PPG-Dalia
>
> Thank you for pointing this out. We ran this additional experiment on the PPG-DaLiA dataset [4], and report the following results. As before, baselines are taken from [1]:
>
> | Model                   | MSE × 10⁻²       |
> |----------------------------|----------------------|
> | NRDE      | 9.90 ± 0.97         |
> | NCDE     | 13.54 ± 0.69        |
> | Log-NCDE  | 9.56 ± 0.59         |
> | LRU      | 12.17 ± 0.49        |
> | S5        | 12.63 ± 1.25        |
> | S6             | 12.88 ± 2.05        |
> | Mamba          | 10.65 ± 2.20        |
> | LinOSS-IMEX                     | 7.5 ± 0.46          |
> | LinOSS-IM                       | 6.4 ± 0.23          |
> | LrcSSM (ours)                         | 10.89 ± 0.96        |
>
> We ran the hyperparameter tuning following the same protocol as before. LrcSSM achieves performance comparable to other models, demonstrating its competitiveness in this benchmark.
>
> > Q8) Moving the pre-normalization block
>
> Our goal was to apply a normalization layer before each SSM block. Moving it outside would only normalize the original input to the first layer, rather than each layer's input. This approach is consistent with other models, where normalization is similarly applied before the SSM layer.
>
> > Q9) Number of layers
>
> We used the same hyperparamater tuning grid as it was used for the other models in [1],[3] to have full comparability. This grid search includes layers={2,4,6}.
>
> > Q10) Loss function for training
>
> We used softmax cross-entropy for the classification tasks.
>
>
> References:
>
> [1] T. Konstantin Rusch and Daniela Rus. "Oscillatory State-Space Models". Thirteenth International Conference on Learning Representations (2025).
>
> [2] Moreno-Pino, Fernando, Alvaro Arroyo, Harrison Waldon, Xiaowen Dong and Álvaro Cartea. “Rough Transformers: Lightweight and Continuous Time Series Modelling through Signature Patching.” Neural Information Processing Systems (2024).
>
> [3] Benjamin Walker, Andrew Donald McLeod, Tiexin Qin, Yichuan Cheng, Haoliang Li, and Terry Lyons. "Log neural controlled differential equations: The lie brackets make a difference." Forty-first International Conference on Machine Learning (2024).
>
> [4] Reiss, Attila, Ina Indlekofer, Philip Schmidt, and Kristof Van Laerhoven. 2019. "Deep PPG: Large-Scale Heart Rate Estimation with Convolutional Neural Networks" Sensors 19, no. 14: 3079. doi: 0.3390/s19143079

---

> > ### Comment · Reviewer_3zsp · 2025-08-04
> > **thanks for the clarifications and additional results**
> >
> > ...

---

> > > ### Author Response · Authors · 2025-08-05
> > >
> > > We appreciate the constructive feedback in the review and the positive response from the reviewer. We are glad that our rebuttal, which aimed to clarify the missing points and included new experimental results, addressed their concerns and overall helped to strengthen our paper.

---

### Official Review · Reviewer_YHNE · 2025-07-03

**Clarity:** 2
**Significance:** 2
**Originality:** 3
**Rating:** 4
**Confidence:** 3

**Summary:**

Inspired by biological neurons, this paper proposes LrcSSM, a nonlinear state space model that yields a diagonal Jacobian matrix. Despite its nonlinearity, the model adopts a linearization-based parallelization strategy using the Jacobian, ensuring numerically stable gradient computation. The proposed approach is validated on both short- and long-horizon time-series classification tasks, demonstrating competitive accuracy compared to baseline models.

**Questions:**

1. Is it appropriate to refer to the nonlinear update as a state-transition matrix, considering that the term conventionally implies linearity in control theory? Wouldn’t this cause confusion?
2. Why are key nonlinear SSM/recurrent baselines such as NSSM, xLSTM, or Quasi-DEER not included in the comparison?
3. Would it be helpful to provide a more detailed explanation of LinOSS when introducing LSSM, given their conceptual similarity?
4. There seems to be some inconsistency in how the proposed model is categorized—it is referred to as both an LSSM and an NSSM in different parts of the paper. Would it be possible to clarify and unify it? It would also be helpful if the authors could explicitly state which class of models their approach belongs to, and highlight in which aspects it offers advantages over existing methods.

**Ethical Concerns:**

["NO or VERY MINOR ethics concerns only"]

**Final Justification:**

While concerns about writing clarity and performance margins remain, the revised perspective on the technical novelty of the proposed approach justifies a more favorable assessment. I am thus raising my score to Weak Accept.

**Limitations:**

yes

**Paper Formatting Concerns:**

No concern for paper formatting

**Quality:**

3

**Strengths And Weaknesses:**

- Strengths
    1. A novel design that clearly separates state-dependent and input-dependent dynamics.
    2. Well-conducted ablation study analyzing both input/state dependency and real-/complex-valued parameterizations.
- Weaknesses
    1. The Introduction contains a lot of awkward wording and phrasing, and suffers from a lack of proper citations. Abbreviations are confusingly defined throughout the paper.
    2. On the UEA benchmark, the proposed model does not show significant performance gains over baseline methods. In particular, it does not clearly outperform LinOSS.
    3. It is unclear whether the authors claim SOTA in  recurrent models. The experimental comparison lacks strong baselines from this category (e.g., xLSTM, Quasi-DEER, other NSSMs).

---

> ### Author Rebuttal · Authors · 2025-07-30
>
> We thank the reviewer for their comments. In the responses below, we aim to clarify the concerns raised.
>
> > 1. State-transition matrix
>
> The reviewer is correct, this term usually refers to linear systems. To incorporate this feedback, we will explicitly highlight that we mean *non-linear* state transition matrix in the text when we talk about non-linear SSMs.
>
> > 2. Comparison with nonlinear RNNs
>
> To the best of our knowledge, there are currently no other NSSMs. However, the reviewer raises an excellent point. To enable comparison with other types of non-linear models, we formulated a general framework that allows scaling any non-linear RNN in the same way as we showed by constructing a diagonal Jacobian.
>
> Using this approach, we extend our method to build NSSMs from Minimal Gated Units (MGU) [1], Gated Recurrent Units (GRU) [2], and LSTM [3], which we refer to as MguSSM, GruSSM and LstmSSM, respectively. We report their performance in the following table:
>
> Test accuracy across *non-linear* models:
>
> | Dataset     | MguSSM      | GruSSM       | LstmSSM      | LrcSSM (ours)      |
> |-------------|---------------------|----------------------|----------------------|----------------------|
> | Heartbeat   | 74.00 ± 4.78        | **75.67 ± 4.67**         | **75.00 ± 3.50**        | **75.0 ± 2.6**           |
> | SCP1        | 78.33 ± 6.62        | 80.24 ± 4.16         | 78.81 ± 3.05         | **84.8 ± 2.8**           |
> | SCP2        | 49.64 ± 9.87        | 52.50 ± 3.31         | 51.07 ± 9.49         | **55.4 ± 7.7**          |
> | EthanolConcentration | 31.05 ± 4.21        | 34.47 ± 1.93         | 32.63 ± 5.61         | **36.1 ± 1.1**           |
> | MotorImagery | **56.43 ± 4.74**        | 49.64 ± 7.27         | 54.29 ± 3.31         | **55.7 ± 4.1**           |
> | EigenWorms  | **90.00 ± 5.15**        | 86.11 ± 6.33         | 82.22 ± 4.51         | 85.6 ± 5.4           |
> | **Average** | 63.2        | 63.1          | 62.34        | **65.4**         |
>
> Here, similar to our ablation studies, we used 6 layers with a 64-dimensional input encoding and 64-dimensional state. The results show that LrcSSM outperforms the other NSSMs built from MGU, GRU and LSTM.
>
> > 3. LinOSS relation to LSSMs
>
> Good point. LinOSS is a special type of LSSM, which is based on forced linear second-order ODEs. Meanwhile, the other LSSMs have first-order linear dynamics.
>
> > 4. Categorization
>
> Thank you for pointing this out. We will go through the paper and clarify the terminology to ensure consistency.
>
> For this and the previous point, we plan to include visualizations to illustrate how these models form categories. However, due to rebuttal restrictions requiring text-only responses, we are unable to provide them at the moment.
>
> In our paper, we refer to NSSMs (*non-linear* state-space models) as those with *non-linear* transition matrices $A$, while LSSMs (linear state-space models) have linear transition matrices, which makes them inherently parallelizable. Our work focuses on the NSSM category, which has traditionally been avoided due to the belief that such models could not be solved in parallel and required sequential processing over the sequence length.
>
> However, recent work [4] introduced the ELK algorithm as a way to parallelize these models efficiently and approximately. Building on this, we propose a model architecture with a diagonal Jacobian structure, enabling exact (not quasi) parallel computation for NSSMs.
>
> Specifically, we explore LrcSSM, a model that leverages this structure alongside the new scaling technique of ELK. As shown in our experiments, LrcSSM performs strongly compared to other NSSMs (see above) and even many leading LSSMs (see the paper).
>
> The key advantage of LrcSSM is that it allows for *non-linear* dynamics within a parallelizable framework, something not achieved before. While this introduces a small number of additional iterations per step, it does not significantly increase runtime and yields performance competitive with, or superior to, strong baselines like LRU (which argued non-linearity was unnecessary), S5, Mamba, and S6.
>
> While LinOSS remains a strong baseline due to its rich second-order dynamics, which we plan to explore further in future work, our LrcSSM model introduces a novel direction by incorporating a *non-linear*, input- and state-dependent transition matrix $A$, which we believe adds a meaningful contribution to the field and opens a new category of exploring NSSMs.
>
> References:
>
> [1] Zhou, Guoxiang, Jianxin Wu, Chen-Lin Zhang and Zhi-Hua Zhou. “Minimal gated unit for recurrent neural networks.” International Journal of Automation and Computing 13 (2016): 226 - 234.
>
> [2] Cho, Kyunghyun, Bart van Merrienboer, Çaglar Gülçehre, Dzmitry Bahdanau, Fethi Bougares, Holger Schwenk and Yoshua Bengio. “Learning Phrase Representations using RNN Encoder–Decoder for Statistical Machine Translation.” Conference on Empirical Methods in Natural Language Processing (2014).
>
> [3] S. Hochreiter and J. Schmidhuber, "Long Short-Term Memory," in Neural Computation, vol. 9, no. 8, pp. 1735-1780, 15 Nov. 1997, doi: 10.1162/neco.1997.9.8.1735.
>
> [4] Xavier Gonzalez, Andrew Warrington, Jimmy Smith, and Scott Linderman. Towards scalable
> and stable parallelization of nonlinear rnns. Advances in Neural Information Processing
> Systems, 37:5817–5849, 2024

---

> > ### Author Response · Authors · 2025-08-05
> > **On the classification of SSMs**
> >
> > We believe the discussion with Reviewer dhK1 already addressed the concerns regarding the terminology and correct categorization of our model.
> >
> > In the revised version of the paper, we will update the definitions and categorization accordingly.
> >
> > We define Non-linear State-Space Models (NSSMs) as models with state-dependent (non-linear) transition matrices, whereas Linear State-Space Models (LSSMs) use transition matrices that are independent of the current state, making them linear in their state dynamics. Both NSSMs and LSSMs can also include input dependencies, which may be either linear or non-linear.
> >
> > From a computational perspective, LSSMs require only a single parallel scan for their forward pass. In contrast, NSSMs rely on the ELK method for parallelization, which typically involves multiple iterations (empirically, around 2-4).
> >
> > More generally, models with internal states, such as traditional nonlinear RNNs, can be considered state-space models (SSMs) in the classical control theory sense. However, in machine learning, the term *structured* SSMs has become more common (often simply referred to as SSMs), where the transition matrix follows a specific structure. This would typically exclude traditional RNNs, which use dense and unstructured transition matrices. In contrast, our LrcSSM, due to its diagonal structure, qualifies as a structured SSM under this definition.
> >
> > We would be happy to hear your thoughts on this, especially if you see alternative perspectives that could further clarify the categorization.

---

> > ### Comment · Reviewer_YHNE · 2025-08-07
> >
> > I appreciate the authors' thorough rebuttal. The clarifications regarding model categorization and the extension to other nonlinear recurrent baselines meaningfully strengthen the contribution. These additions help position LrcSSM more clearly within the broader landscape of NSSMs and demonstrate its relative advantages.

---

> > > ### Author Response · Authors · 2025-08-07
> > >
> > > We sincerely thank the reviewer for their feedback and for taking the time to review our rebuttal. We are glad that the clarifications and additional comparisons helped strengthen the positioning of our LrcSSM model and contributed to improving the paper.

---

### Official Review · Reviewer_dhK1 · 2025-07-10

**Clarity:** 2
**Significance:** 3
**Originality:** 3
**Rating:** 5
**Confidence:** 5

**Summary:**

This paper introduces LrcSSM, a switching linear system (given by equations 12-14), where the diagonal transition matrix A depends on both input u **and state x** (i.e., the system is still nonlinear), They show that LrcSSM can outperform Mamba on three long time series classification tasks  drawn from sensor recordings, the longest of which (T=18K) is eigenworms.

**Questions:**

The actionable items that I would like to see to give this paper an accept center on the experiments.

1. I need to see Long Range Arena (LRA) (https://arxiv.org/abs/2011.04006) included as a second experiment to advocate for publication of this paper. I don't know if LrcSSM will do better than LRU on this dataset or not---and I don't think it's critical for publication that LrcSSM do better on LRA---but I think it's very important for publication for this benchmark dataset to be included. LRA is much more standard than UEA-MTCSA and I think it's really important for readers to be able to see the results. If LrcSSM crushes on LRA---great!! If it has weaker performance, I think this opens up a great opportunity to dive deep into the differences between these datasets and comment on use cases for linear vs nonlinear RNNs.
2. The abstract explicitly claims that the model follows compute-optimal scaling law regime. However, the evidence for this is give only in appendix A.2, which states
> Hence, $\beta \approx 0.42$ is a defensible prior for LrcSSM
To me, it seems like a claim made in the abstract should be demonstrated empirically or LrcSSM itself, not handwaved in the appendix based on mamba experiments.
3. Something is odd with the ablations reported in Table 7. What are the $\pm$ terms and how are they computed? And why are they so much higher for average than for the original experiments. It seem like based on the very large $\pm$ terms, the differences between the different architectures are not statistically significant---is this a correct interpretation? In essence, it looks like input dependent only is on par with state dependent (and will run faster as it actually will converge in the 1 pscan). Could the authors comment on this interpretation of Table 7?
4. Would it be possible to report the wallclock times for the different architectures? I'm curious how their training time compares.
5. Would it be possible to add Liquid-S4 to the baselines? It is strange that Liquid-S4 is mentioned by name in the abstract, but not included in the experiments.
6. Do we really believe the mamba performance on eigenworms? it seems unbelievably low, as if a real effort on getting strong performance was not made. I see that it is reported from another paper, but would it be possible to try to train your own mamba baseline, to genuinely try to make it as strong as possible, and to document your approach? For example, in Table 7, in the input dependent only (not state dependence) LrcSSM is reported at 86.7% on eigenworms. Mamba, which is also input dependent, is reported at 70%. How can this be? Can the authors comment on why there is such a big difference between these two input dependent switching linear systems? This is extremely hard for me to believe, I would have thought that Mamba and input-dependent-only LrcSSM would be extremely similar.
7. Could the authors clarify whether they use ELK or quasi-DEER in their experiments? If they use ELK, how did they set the hyperparameter $\lambda$, and could they report its value?
8. Would it be possible to include a fair comparison against a transformer baseline, such as in Amos et al "Never train from scratch" (https://arxiv.org/pdf/2310.02980)

**Ethical Concerns:**

["NO or VERY MINOR ethics concerns only"]

**Final Justification:**

I raised my score because of the strengthened experimental evaluation. In particular, their wall clock time results are extremely exciting, indicating the speed of LrcSSM. Also exciting are the fast convergence rates of quasi-DEER. These results indicate that LrcSSM has tremendous promise to work at large scale.

This innovative paper has opened up new perspectives for me on the parallelization of nonlinear recurrent architectures. It has already begun to affect how I think about future research directions. Papers like these should definitely be discussed at conference like NeurIPS. I have strong conviction and confidence that this paper should be accepted for publication.

**Limitations:**

The abstract appears to contradict the limitations section.

The abstract reads (lines 2-4)
> By forcing the state-transition matrix to be diagonal and learned at every step, the full sequence can be solved in parallel with a single prefix-scan.

Limitations section says (lines 268-9)
> However, we also have to take into account that LRCs solved by ELK needed more Newton steps to converge at each iteration, which linear SSMs do not require.

For publication, I would really like to see graphs/tables showing the number of Newton steps needed for LrcSSM to converge, and a comparison of wall clock time (to answer the question: is LrcSSM much slower in practice than its linear counter parts?).

**Paper Formatting Concerns:**

Formatting is great! I like the use of color!

Line 178: you don't need to say "due to obvious space limitations." Can just cut this.

**Quality:**

2

**Strengths And Weaknesses:**

Strengths:
* Very clever way to make quasi-DEER and quasi-ELK work at scale---simply make the nonlinear RNN diagonal!
* interesting choice of benchmark with UEA-MTCSA, using real-world measurements of dynamical systems

Weaknesses:
* Limited experimental validation: the experiments are pretty slim and should be built out
* Based on this paper, I really don't know if this approach works at scale any better than existing approaches

---

> ### Author Rebuttal · Authors · 2025-07-30
>
> We thank the reviewer for their highly constructive feedback. We appreciate it a lot and will incorporate it into our revision. Here are our replies:
>
> > 1. LRA benchmarks and comparing linear vs. non-linear models
>
> Thank you for the suggestion. We would like to point out that, particularly in recent work, researchers have increasingly shifted from the LRA benchmarks to the UEA-MTSCA archive. Our decision to focus on UEA-MTSCA aligns with this trend, as reflected in several recent top publications [1]-[3]. To further support this direction, we also cite here additional examples published on arXiv after the NeurIPS submission, in May/June [4]-[8].
>
> That said, we completely agree that including LRA would provide valuable context and comparison. We are very interested in exploring how our non-linear LrcSSM performs on LRA, especially in relation to recent linear models such as LinOSS, which also don't report performance on these benchmarks. However, we believe such a broader investigation would be more out of the scope of the current paper.
>
> Nonetheless, one can address this point in future work by exploring which types of datasets benefit more from Transformer-like, linear-SSM-like, or non-linear-SSM-like architectures.
>
> For now, as also pointed out by other reviewers, we ran additional experiments on the PPG-DaLiA dataset [9], and can report the following results (baselines are taken from [3]):
>
> | Model                   | MSE × 10⁻²       |
> |----------------------------|----------------------|
> | NRDE      | 9.90 ± 0.97         |
> | NCDE     | 13.54 ± 0.69        |
> | Log-NCDE  | 9.56 ± 0.59         |
> | LRU      | 12.17 ± 0.49        |
> | S5        | 12.63 ± 1.25        |
> | S6             | 12.88 ± 2.05        |
> | Mamba          | 10.65 ± 2.20        |
> | LinOSS-IMEX                     | 7.5 ± 0.46          |
> | LinOSS-IM                       | 6.4 ± 0.23          |
> | LrcSSM (ours)                         | 10.89 ± 0.96        |
>
> We ran the hyperparameter tuning following the same protocol as before. LrcSSM achieves performance comparable to other models, demonstrating its competitiveness in this benchmark.
>
> >2. Compute-optimal scaling law regime
>
> Thank you for raising this point. We agree that claims in the abstract should be clear and well-supported. Our intention was not to present a new empirical measurement of the scaling exponent $\beta$ for LrcSSM, but rather to argue based on properties of LrcSSM that it is expected to follow the same compute-optimal scaling regime. We will revise the abstract to clarify this and note that a full scaling study for LrcSSM is a valuable direction for future work.
>
> >3. Table 7 and input dependency
>
> In Table 7, we report the mean and standard deviation of accuracy across the 5 different dataset splits. We observed that performance can vary substantially from split to split, which naturally leads to higher standard deviations.
>
> The deviations appear larger in Table 7 compared to earlier tables because these results are not fully tuned over the entire hyperparameter grid. Instead, we use a standardised configuration across ablations (6 layers, 64-dimensional input encoding, and 64 hidden states) to ensure comparability across the model variants.
>
> Regarding statistical significance: yes, the standard deviations are relatively large, which makes the differences between variants less statistically robust. However, this is a common observation in Table 3 as well, where many baseline models show similar variance. This stems from the inherent variability in achievable accuracies across the datasets.
>
> Despite the high variance, we focused on identifying configurations with the highest mean accuracy. Your interpretation is correct: the input-dependent variant, which converges in 1 iteration, but performs slightly worse than the state-dependent version. Given this trade-off, we choose the better performing model, as it does not impose a significant computational overhead, please check our reported results in the next point.
>
> > 4. Wall-clock times
>
> Wall clock time (in seconds) for 1000 training steps of the best performing (tuned) models for each dataset, same as it was reported in [3].
>
> | Dataset     | NRDE  | NCDE  | Log-NCDE | LRU | S5 | Mamba | S6 | LinOSS-IMEX | LinOSS-IM | LrcSSM (ours) |
> |-------------|-------|-------|----------|-----|----|--------|-----|--------------|------------|--------|
> | Heartbeat   | 9539  | 1177  | 826      | 8   | 11 | 34     | 4   | 4            | 7          | 24     |
> | SCP1        | 1014  | 973   | 635      | 9   | 17 | 7      | 3   | 42           | 38         | 12     |
> | SCP2 | 1404  | 1251  | 583      | 9   | 9  | 32     | 7   | 55           | 22         | 16     |
> | EthanolConcentration | 2256  | 2217  | 2056     | 16  | 9  | 255    | 4   | 48           | 8          | 17     |
> | MotorImagery | 7616  | 3778  | 730      | 51  | 16 | 35     | 34  | 128          | 11         | 29     |
> | EigenWorms  | 5386  | 24595 | 1956     | 94  | 31 | 122    | 68  | 37           | 90         | 33     |
>
> Iterations needed for convergence for LrcSSM on average per dataset in the same order: 3.4, 2.1, 4.4, 4.6, 3.0, 3.8.
> To make it explicit, this number is 1.0 for all other models across all datasets.
> As one can see, we do not lose significant runtime due to the extra iterations that are needed for the states to converge.
>
> > 5. Liquid-S4 baseline
>
> Instead of reporting results for Liquid-S4 directly, we include S6, which builds on the same core idea, which is an input-dependent
> $A$ matrix, but incorporates parallel scan operations for improved efficiency. In this sense, S6 can be viewed as a more efficient version of Liquid-S4, and we include it as a representative of that model class.
>
> > 6. Mamba results
>
> We would like to clarify that Mamba is an overall architecture design that uses S6 (or other SSM layers) as its core building block. Therefore, the more appropriate comparison is between S6 and LrcSSM. For the Eigenworms dataset, we report an accuracy of 85.0% for S6, which aligns closely with our results of the input-dependent version of LrcSSM in Table 7.
>
> The difference in performance between S6 and the input-dependent-only LrcSSM likely stems from the linear nature of S6 vs the non-linear model of LrcSSM. This distinction can explain the performance gap observed despite both being input-dependent systems.
>
> >7. ELK
>
> Thank you for pointing this out. We use (quasi)-ELK, which in our case is equivalent to ELK because our Jacobian matrices are diagonal. Thus, it is not an approximation but an exact computation for our model. We used the default parameter setting, with lambda=1/10^8. We will include this information explicitly in the paper.
>
> >8. Comparison against Transformers
>
> Good point. Before, we missed this recent publication from last NeurIPS, which introduced a new Transformer model called Rough Transformer (RFormer) [1] and evaluated it on the same benchmarks. LrcSSM achieves better performance compared to the Transformer and competitive/better than the Rough Transformer baselines across the evaluated datasets. We will incorporate these results into our paper.
>
> | Dataset    | Transformer       | RFormer          | LrcSSM (ours)     |
> |------------|-------------------|------------------|-------------------|
> | Heartbeat  | 70.5 ± 0.1        | **72.5 ± 0.1**       | **72.7 ± 5.7**        |
> | SCP1 | 84.3 ± 6.3        | 81.2 ± 2.8       | **85.2 ± 2.1**        |
> | SCP2 | 49.1 ± 2.5        | 52.3 ± 3.7       | **53.9 ± 7.2**       |
> | EthanolConcentration | **40.5 ± 6.3**        | 34.7 ± 4.1       | 36.9 ± 5.3        |
> | MotorImagery | 50.5 ± 3.0        | 55.8 ± 6.6       | **58.6 ± 3.1**        |
> | EigenWorms | OOM               | **90.3 ± 0.1**       | **90.6 ± 1.4**        |
> | **Average**| 59.0       | 64.5      | **66.3**       |
>
> > Limitations:
> - What we meant by "parallel with a single prefix-scan" is that each iteration of our iterative algorithm processes the entire sequence of length $T$ simultaneously, rather than processing the sequence elements one at a time as in non-linear RNNs. We'll revise our text to make this clear.
> - See point 4.
>
> References:
>
> [1] Moreno-Pino, Fernando, Alvaro Arroyo, Harrison Waldon, Xiaowen Dong and Álvaro Cartea. “Rough Transformers: Lightweight and Continuous Time Series Modelling through Signature Patching.” Neural Information Processing Systems (2024).
>
> [2] Benjamin Walker, Andrew Donald McLeod, Tiexin Qin, Yichuan Cheng, Haoliang Li, and Terry Lyons. "Log neural controlled differential equations: The lie brackets make a difference." Forty-first International Conference on Machine Learning (2024).
>
> [3] T. Konstantin Rusch and Daniela Rus. "Oscillatory State-Space Models". Thirteenth International Conference on Learning Representations (2025).
>
> [4] Walker, Benjamin, Lingyi Yang, Nicola Muca Cirone, Cristopher Salvi, and Terry Lyons. "Structured Linear CDEs: Maximally Expressive and Parallel-in-Time Sequence Models." arXiv preprint arXiv:2505.17761 (2025).
>
> [5] Boyer, Jared, T. Konstantin Rusch, and Daniela Rus. "Learning to Dissipate Energy in Oscillatory State-Space Models." arXiv preprint arXiv:2505.12171 (2025).
>
> [6] Pourcel, Guillaume, and Maxence Ernoult. "Learning long range dependencies through time reversal symmetry breaking." arXiv preprint arXiv:2506.05259 (2025).
>
> [7] Nzoyem, Roussel Desmond, Nawid Keshtmand, Idriss Tsayem, David AW Barton, and Tom Deakin. "Weight-Space Linear Recurrent Neural Networks." arXiv preprint arXiv:2506.01153 (2025).
>
> [8] Karuvally, Arjun, Franz Nowak, Anderson T. Keller, Carmen Amo Alonso, Terrence J. Sejnowski, and Hava T. Siegelmann. "Bridging Expressivity and Scalability with Adaptive Unitary SSMs." arXiv preprint arXiv:2507.05238 (2025).
>
> [9] Reiss, Attila, Ina Indlekofer, Philip Schmidt, and Kristof Van Laerhoven. 2019. "Deep PPG: Large-Scale Heart Rate Estimation with Convolutional Neural Networks" Sensors 19, no. 14: 3079. doi: 0.3390/s19143079

---

> ### Comment · Reviewer_dhK1 · 2025-08-01
> **I believe this paper should be accepted**
>
> I congratulate the authors on their great work and well-considered reviews. I am particularly heartened by their strengthened experimental evaluation. In particular, their wall clock time results are extremely exciting, indicating the speed of LrcSSM. Also exciting are the fast convergence rates of quasi-DEER. These results should definitely be included in the final paper.
>
> Can the authors provide richer distributional statistics for the iterations needed for convergence for LrcSSM? That is, instead of just average, can they provide min, Q1, median, Q3, and max? (should be doable with one line of code, `df.summary()`). Doing so would give a better perspective as to possible worst case behavior during training and pathologies.
>
> On the balance, this innovative paper has opened up new perspectives for me on the parallelization of nonlinear recurrent architectures. It has already begun to affect how I think about future research directions. Papers like these should definitely be discussed at conference like NeurIPS. I have strong conviction and confidence that this paper should be accepted for publication. I have raised my score to a "5."
>
> --------
>
> I include some additional feedback aimed at further strengthening the paper. I have three minor points from this review, and a major point from Reviewer YHNE.
>
> > Table 7
>
> I would recommend reporting the standard *error* instead of the standard *deviation*. This way, the average would illustrate a lower standard error (as it must), instead of a standard deviation that is larger than any particular experiment. Doing so would be more statistically accurate, and also would strengthen your story.
>
> > Mamba on Eigenworms
>
> I am still underwhelmed by your discussion of the performance of Mamba on eigenworms. S6 is a part of Mamba. Presumably the authors of Mamba would not have made additions to from S6 to Mamba if they did not believe that improved performance. Can you please comment on the differences between S6 and Mamba, and why you believe that S6 is outperforming Mamba so dramatically here?
>
> > quasi-DEER or quasi-ELK
>
> I looked through your very well organized code. It appears to me that quasi-DEER, and not quasi-ELK, is being called. Can you point me to the line in your zip code that calls quasi-ELK in your model training architecture?
>
> # Point from reviewer YHNE: what is a NSSM?
>
> I am confused by your discussion of NSSM.
>
> > In our paper, we refer to NSSMs (non-linear state-space models) as those with non-linear transition matrices, while LSSMs (linear state-space models) have linear transition matrices, which makes them inherently parallelizable.
>
> Both LrcSSM and Mamba are switching linear dynamical systems. Mamba's transition matrix is also a nonlinear function of its inputs. The difference between LrcSSM and Mamba is what their transition matrices are functions *of*. Mamba is only input dependent, which is why it can be parallelized in a single parallel scan. LrcSSM is *state*-dependent, which is why it requires quasi-DEER for parallelization.
>
> I also do not think that reserving the term "SSM" for (switching) *linear* dynamical systems is a good decision. A "state space model" has classically referred to *any* stateful (Markovian) system, with arbitrary transition function. Under this more traditional definition, Reviewer YHNE is correct that xLSTM is a nonlinear state space model (because of the nonlinear dynamics in sLSTM). I agree with their recommendation that xLSTM should be added to the experiments (including wall clock time).

---

> ### Author Response · Authors · 2025-08-01
>
> We thank the reviewer again for their thoroughness, including reading the other reviewers' comments, and for recognizing the value of our contribution by raising the score.
>
> Below, we provide our responses:
>
> > Statistics for the iterations in LrcSSM
>
> | Dataset      | Min | Q1   | Median | Q3   | Max   |
> |--------------|-----|------|--------|------|-------|
> | Heartbeat    | 3   | 3    | 3      | 4    | 5.25  |
> | SCP1         | 2   | 2    | 2      | 2    | 4     |
> | SCP2         | 3   | 4    | 4      | 5    | 7.25  |
> | EthanolConcentration      | 3   | 3    | 4.75   | 6    | 6     |
> | MotorImagery        | 3   | 3    | 3      | 3    | 3     |
> | EigenWorms   | 3   | 3.75 | 4      | 4    | 4     |
>
> We will provide this in a boxplot format in the paper.
>
> > Table 7
>
> Thank you for the suggestion. We agree that standard error is more appropriate, as it better reflects confidence in the mean than standard deviation.
>
> Following your recommendation, we will revise Table 7 to report standard errors instead. For reference, the updated values are:
> 65.4 ± 8.02, 65.0 ± 8.04, 63.8 ± 8.37
>
> We will also apply this adjustment consistently across all relevant tables in the paper.
>
> > Mamba on EigenWorms
>
> We agree that the reviewer raises a valid point and appreciate the opportunity to further share our thoughts. Our understanding is that the additions introduced in Mamba were designed to improve performance on the specific datasets evaluated in the Mamba paper. However, in the context of the tasks evaluated here, particularly the Eigenworms dataset, these enhancements do not appear to offer the same benefit.
>
> It is important to note that the results we report for both S6 and Mamba come from a different paper [1], not the original Mamba paper [2]. While we did not reproduce these results ourselves, we assume they are valid and reported in good faith. Nonetheless, the observed performance gap between S6 and Mamba on this dataset remains a task for future investigation.
>
> > quasi-DEER or quasi-ELK
>
> Thank you for catching this, the reviewer is absolutely right. This is a mistake on our part. We used quasi-DEER, which is in fact a specialization of quasi-ELK when the lambda values are close to zero. We will correct the terminology in the paper to accurately reflect this, and we appreciate you pointing it out.
>
> > NSSM discussion, dependency and (non)-linearity
>
> We agree with the reviewer on the key distinction regarding dependency: Mamba is input-dependent, while LrcSSM is both input- and state-dependent in $A$, this is well understood.
>
>
> We would like to clarify some aspects regarding parallel computations.
>
> Mamba uses linear projections [2] and includes exponential terms for discretization for $\bar A$, where the time-varying component comes from input dependency. The Mamba authors were able to construct the computations in an associative manner, with a customized parallel scan. However, this approach does not generalize to arbitrary non-linear projections in $A$. In the general case, methods like ELK are needed to achieve parallelization.
>
> Overall, when we refer to non-linear SSMs, the non-linear nature is a core part of $A$.
>
> Regarding xLSTM [3], including experiments on it is a valuable suggestion. However, due to time constraints, we are unable to provide these during the rebuttal period. We hypothesize that its performance is reflected in our LstmSSM ablation study, which we have included in the rebuttal.
>
> [1] Benjamin Walker, Andrew Donald McLeod, Tiexin Qin, Yichuan Cheng, Haoliang Li, and Terry Lyons. "Log neural controlled differential equations: The lie brackets make a difference." Forty-first International Conference on Machine Learning (2024).
>
> [2] Gu, Albert, and Tri Dao. "Mamba: Linear-time sequence modeling with selective state spaces." arXiv preprint arXiv:2312.00752 (2023).

---

> ### Comment · Reviewer_dhK1 · 2025-08-01
> **Misunderstanding about parallelizability**
>
> ## Statistics for the iterations in LrcSSM
>
> These are excellent! Well done! I agree their inclusion makes the paper stronger.
>
> ## quasi-DEER or quasi-ELK
>
> Yes, quasi-DEER is quasi-ELK with $\lambda=0$ (no regularization). However, you are right to directly use the quasi-DEER code for numerical and computational reasons.
>
> ## When you can parallelize a linear dynamical system (LDS)
>
> We appear to have a fundamental misunderstanding about which LDSs can be parallelized. Luckily, the results of the paper hold as is. However, I believe that clearing up this misunderstanding is important for contextualization of this paper.
>
> I believe that any LDS with transition matrix NOT dependent on state can be parallelized. This includes an LDS with transition matrices that depend on the inputs in nonlinear ways. I hope that we can discuss this point from first principles.
>
> Blelloch ('91) has written an manuscript on the parallel associative scan, also known as the prefix sum operator. This parallel scan is implement in jax as `jax.lax.associative_scan` and is used by your code. The parallel scan parallelizes matrix multiplication with a binary tree like approach.
>
> I can write in pseudocode a way to parallelize an LDS with $A_t$ depending nonlinearly on inputs $u_t$.
>
> Say $A_t = f(u_t)$, where $f$ is an arbitrary (possibly nonlinear) function.
>
> 1. `As = jax.vmap(f)(us)`, where `us` are all the inputs (must be known in advance, otherwise we could not parallelize. This operation is embarrassingly parallel. `As` is of shape `(T,D,D)`, and is the transition matrices (depending nonlinearly on `u_t`).
> 2. `jax.lax.parallel.scan(matmul_op, As)`  gives the output in a parallel manner, running in $O(\log (T))$ time.
>
> This fact that *any* input dependent switching LDS is parallelizable with one applicaition of a parallel associative scan is best explained in Feng et al, 2024, "Were RNNs All we needed." See their beautiful algorithm blocks for the minGRU and minLSTM and their excellent background section, particularly section 2.3
>
> I hope that the authors can clear up this misunderstanding, because doing so is important for contextualizing the paper!

---

> > ### Author Response · Authors · 2025-08-01
> >
> > We thank the reviewer for their responsiveness and agree that the discussion below does not affect the validity of our models or results. Nonetheless, we appreciate and value this exchange.
> >
> > We acknowledge that we may have gotten a bit stuck on what exactly constitutes "non-linearity" in the transition matrix, which diverted our earlier points. We believe we can now come to a clearer common ground.
> >
> > Input dependency and parallelization:
> >
> > We agree with the reviewer that, regardless of whether the transition matrix is linear or non-linear in its input, if the matrix is *only* input-dependent, it can be parallelized via a parallel scan, as the reviewer correctly pointed out.
> >
> > Mamba's transition matrix:
> >
> > We still think Mamba's transition dynamics are somewhat nuanced. Technically, $A$ itself is not directly input-dependent, but due to its interaction with $\Delta$ during discretization, the effective transition becomes input-dependent. The authors also suggest that $A$ could be made input-dependent independently of $\Delta$. In that case, whether the transition matrix is linear or non-linear in its input would depend on the form of this dependency in $A$.
> >
> > - This raises a broader question: if discretization introduces an exponential term that involves the input, should all input-dependent transition matrices be considered non-linear in their inputs due to their discrete form?
> > - In the case of Liquid-S4, the transition matrix would be linearly dependent on the input with a bias term.
> > - For comparison, in the minGRU and minLSTM models mentioned in the paper, the transition matrices are non-linear in their input. This case is much more explicit.
> >
> > Overall, when highlighting models with input-dependent dynamics, it remains difficult to draw a clear distinction between linear and non-linear input dependencies. Previously, we considered all input-dependent transition matrices of SSMs in the literature to be linear in the input, a point that, in hindsight, is not so clear (minGRU is non-linear, but is it an SSM?). This leads us to our final point below.
> >
> > To clarify our earlier sentence, which, in hindsight, lacked precision and may have created this longer (but valuable) discussion:
> > > In our paper, we refer to NSSMs (non-linear state-space models) as those with non-linear transition matrices, while LSSMs (linear state-space models) have linear transition matrices, which makes them inherently parallelizable.
> >
> > NSSMs have (non-linear) state-dependent transition matrices, while LSSMs use transition matrices without state dependency. In other words, NSSM systems are non-linear in their state, while LSSMs are linear. Both types of systems can also have input dependency, which may be either linear or non-linear.
> >
> > Please let us know if we are now aligned on the categorization and terminology.

---

> > > ### Comment · Reviewer_dhK1 · 2025-08-02
> > > **Definition of SSM**
> > >
> > > Thank you for engaging in this productive conversation! I believe we have strengthened the paper by clarifying its context.
> > >
> > > ## Definition of SSM
> > >
> > > I think the core definition is that of a state-space model, or SSM.
> > >
> > > I said the following in my first comment:
> > >
> > > > I also do not think that reserving the term "SSM" for (switching) linear dynamical systems is a good decision. A "state space model" has classically referred to any stateful (Markovian) system, with arbitrary transition function. Under this more traditional definition, Reviewer YHNE is correct that xLSTM is a nonlinear state space model (because of the nonlinear dynamics in sLSTM). I agree with their recommendation that xLSTM should be added to the experiments (including wall clock time).
> > >
> > > I still agree with my assessment. SSM has traditionally referred to any model that has a *state*. This is the definition that is used in the `dynamax` package that is used, for example, in the ELK codebase that you include in your zipped code.
> > >
> > > Under this definition, SSMs include more traditional nonlinear RNNs like LSTMs and GRUs. In this way, LrcSSM is not the first nonlinear SSM---it instead builds on a long line of work including the LSTM, the GRU, the sLSTM, and many more. Thus, I believe your comment "To the best of our knowledge, there are currently no other NSSMs" to reviewer YHNE is not helpful and would be confusing to readers.
> > >
> > > However, this is not a big deal, because you correctly scoped your contributions in the main text of your manuscript (lines 45-52).
> > >
> > > The key word is "bio-inspired," which Lrc absolutely is. I think you could also point out that LrcSSM is novel in considering a switching LDS where the transition matrix $A_t$ also has *state* dependence. Therefore, I still believe your contributions are very significant and exciting! Moreover, as I stated in my review, the diagonal architecture was very creative and, in the nonlinear setting, novel to my knowledge.
> > >
> > > I still believe the right distinction between "nonlinear" and "linear" SSMs depends on how nonlinear they are with respect to the state, which also distinguishes how they can be parallelized.
> > >
> > > "linear" SSMs like mamba evolve their state linearly, according to $x_{t+1} = A_t x_t$. This does not preclude $A_t$ from depending nonlinearly on inputs or parameters, but crucially $A_t$ is NOT a function of $x_t$. Therefore, in the update $x_{t+1} = f(x_t)$, the transition function $f(x_t) = A_t x_t$ is a *linear* function of $x_T$. Consequently, linear SSMs like mamba require only one parallel scan for their forward pass.
> > >
> > > "nonlinear" SSMs like GRU, LSTM, sLSTM, or LrcSSM evolve their state nonlinearly, according to $x_{t+1} = f(x_t)$, where $f(x_t)$ is not a linear function. LrcSSM parameterizes $f(x_t) = A(x_t, u_t) x_t$. But, crucially, by making $A$ dependent on $x_t$ (note this dependence could have been linear or nonlinear, that point is irrelevant), the transition function $f$ becomes a nonlinear of the state $x_t$. Consequently, such nonlinear SSMs require DEER to parallelize their forward pass, which in general requires more than one parallel scan.
> > >
> > > ### Discrete vs continuous time is a red-herring
> > >
> > > I want to point out that a criticism of traditional nonlinear RNNs is that they appear to be operating in discrete time. However, they can always be treated as a discretization of a continuous time system.
> > >
> > > Consider the discrete time recurrent model
> > >
> > > $ x_{t+1} = f(x_t)$.
> > >
> > > Note that this can always be written as an Euler discretization of the continuous time recurrent model
> > >
> > > $ \dot{x} = -x + f(x),$
> > >
> > > using step size $\Delta t = 1$.
> > >
> > > Therefore, any of these models could also be used with varying discretization parameters (i.e. $\Delta t \neq 1$).

---

> > > > ### Author Response · Authors · 2025-08-03
> > > >
> > > > We thank the reviewer again for their thoughtful discussion. Many of the discussed points are insightful and valuable beyond this exchange, and we plan to include them in the appendix and/or as part of an extended knowledge base.
> > > >
> > > > We believe we are now fully aligned on the terminology. However, there is one additional distinction we would like to clarify regarding SSMs. As the reviewer pointed out, many models with states, such as traditional non-linear RNNs, can indeed be viewed as SSMs in the classical sense from control theory. In machine learning, however, the notion of *structured* SSMs has gained popularity (but many times just called SSMs), where the transition matrix $A$ has a specific structure. This would typically exclude traditional RNNs due to their dense and unstructured transition matrices, whereas LrcSSM would qualify as a structured SSM due to its diagonal structure.
> > > >
> > > > We also agree with the reviewer that removing the sentence *"To the best of our knowledge, there are currently no other NSSMs"* is a good idea. This will avoid unnecessary confusion, since, as the reviewer also pointed out, we already highlight the unique aspects of our approach, such as bio-inspiration and non-linear state- and input-dependence.
> > > >
> > > > Regarding traditional RNNs, the continuous-time formulation is an excellent point and aligns with our line of thought perfectly. In general (beyond this paper), we also believe this perspective deserves wider recognition.
> > > >
> > > > Thank you again for the engaging discussion.

---

### Author Response · Authors · 2025-08-08
**Reply to all Reviewers - Summary of feedback and improvements**

As we approach the end of the discussion period, we would like to sincerely thank the reviewers once again for their high-quality and constructive feedback. The updated revised paper will incorporate the reviewers' suggestions. In summary, the rebuttal and discussions have led to the following improvements:

- Reporting wall-clock times and number of iterations of our method

- Clearer categorization within the SSM landscape. We especially appreciate the engaging discussion with Reviewer dhK1, which helped refine the categorization of our model within SSMs.

- Generalization of our approach to any kind of non-linear RNN (building a model with diagonal Jacobian, enabling parallelization with exact computations instead of quasi-approximations) to demonstrate broader impact

- Additional comparisons and ablation studies: comparison against Transformers, non-linear models, bio-inspired models, additional experiment on PPG-DaLiA, comparison between the non-simplified model (dense Jacobian) and our version (diagonal Jacobian)

- Additional clarifications to be included in the paper

We sincerely *thank all reviewers for their positive feedback and for expressing satisfaction with our rebuttal*. We are glad that our clarifications, additional experiments, and extended comparisons addressed their questions and collectively helped strengthen the paper.

---

### Decision · Program_Chairs · 2025-09-17

**Decision:**

Accept (poster)

**Comment:**

The authors present a novel deep SSM architecture that has nonlinear dynamics within each layer. The authors leverage recent advances in parallelization of nonlinear recursions to render their architecture computationally tractable. As Reviewer dhK1 said,
> This innovative paper has opened up new perspectives for me on the parallelization of nonlinear recurrent architectures. It has already begun to affect how I think about future research directions. Papers like these should definitely be discussed at conference like NeurIPS.

I agree with this assessment and I congratulate the authors on their nice work.